# Predicting customer loyalty in omnichannel retailing using purchase behavior, socio-cultural factors, and learning techniques

**Shima Roosta**[ORCID]*, **Seyed Jafar Sadjadi**\*, **Ahmad Makui**

Faculty of Industrial Engineering, Iran University of Science and Technology, Tehran, Iran

\* shima.roosta@yahoo.com (SR); sjsadjadi@iust.ac.ir (SJS)

## Abstract

In the competitive retail omnichannel market, customer loyalty is essential for maintaining market share and reducing the cost of acquiring new customers. Previous research has primarily focused on factors influencing customer loyalty, often in isolation, but this study goes beyond traditional approaches. The aim of this research is to fill significant gaps in current studies by integrating a more comprehensive set of variables that reflect the complex and dynamic nature of customer loyalty in a flexible omnichannel environment. The main innovation of this study lies in the use of new and comprehensive omnichannel data, which includes sales data across various platforms, socio-economic conditions, shopping cart behaviors, and customer sentiments. The proposed model utilizes a hybrid approach, incorporating BERT for sentiment analysis, reinforcement learning for behavior analysis, and fine-tuning for improved predictions. Additionally, graph-based models (GCN) and adaptive learning are employed to analyze trends and predict loyalty at both individual and neighborhood levels. This research provides an intelligent analytical framework for predicting customer loyalty in omnichannel retail environments, enhancing Customer Relationship Management (CRM) subsystems within Enterprise Information Systems (EIS). By optimizing decisions in areas such as pricing, inventory management, and personalized advertising, this study ultimately leads to improved customer retention and increased market competitiveness.

## 1. Introduction

With the expansion of e-commerce and increasing market competition, customer expectations are continuously rising, and they demand a flawless shopping experience [1]. Omnichannel strategies in the retail industry play a crucial role in meeting these expectations by integrating various channels (from physical stores to websites and mobile applications) and providing a seamless customer experience [2,3]. As technology advances and consumer behavior evolves, omnichannel strategies are

**Data availability statement:** The dataset used in this study is available on Kaggle at the following link: [https://www.kaggle.com/datasets/bhavikjikadara/retail-transactional-dataset?resource=download]. The data can be accessed upon reasonable request. If any ethical, privacy, or security concerns arise, access to the data may be restricted accordingly.

**Funding:** The author(s) received no specific funding for this work.

**Competing interests:** The authors have declared that no competing interests exist.

increasingly attracting the attention of traditional retailers and have become a popular strategy among them. Research particularly indicates that implementing omnichannel strategies not only enhances customer satisfaction but also contributes to customer loyalty and the acquisition of new customers [1–6].

In today's competitive world, brands that are able to create loyalty among their customers not only succeed in attracting new customers, but also reduce marketing costs, while establishing more sustainable and profitable relationships with their customers and acting more effectively in selling their products and services. This is particularly important in the digital world and for businesses that have an omnichannel strategy, especially omnichannel retailers [7]. One way to build an effective relationship with customers is to understand customer satisfaction and loyalty. Considering the numerous studies published on customer loyalty across various sectors, it is clear that the importance of loyalty has become increasingly evident for businesses. Among these studies, comprehensive research has been published [8–13]. Interested readers can refer to these sources for further insights. Customer loyalty refers to the factors that influence customer behavior to ensure continuous and long-term purchases. Research shows that retaining existing customers is significantly more cost-effective than acquiring new ones [14]. Therefore, organizations must find ways to establish lasting relationships with their customers, strengthen their loyalty, and ensure customer retention. Loyal customers frequently purchase from a business and may even recommend it to others [15]. Given the importance of customer loyalty and its prediction, identifying and analyzing the factors influencing loyalty is essential.

To better understand the practical application of customer loyalty prediction models, several studies have examined loyalty in major brands and multichannel retailers. One notable study involved predicting loyalty in a telecommunications company using demographic data and a novel TFM approach [16]. Other practical examples include research studies [8,17,18] that have explored customer loyalty prediction in hotels, banks and online fashion stores using real-world case studies. Various studies indicate that customers' purchasing decisions and consequently their loyalty are influenced by a combination of economic, social, cultural, and emotional factors, which are examined below. Generally, purchasing decisions are shaped by multiple influences. According to Hofstede's Cultural Dimensions Theory, cultural factors significantly impact customers' buying decisions. For instance, while collectivist cultures place greater importance on social recommendations, individualistic cultures tend to prefer faster, more independent purchasing decisions [19–22]. Another crucial factor influencing purchasing decisions is customer emotions, which aligns with Oliver's (1999) customer loyalty model. This model views loyalty as a gradual process, where customers first become aware of a brand through cognitive recognition, then develop emotional and behavioral loyalty through positive experiences. Ultimately, in the action phase, loyal customers consistently purchase from the brand and recommend it to others [23,24].

Additionally, social influences play a significant role in customer loyalty, as explained by Ajzen's Theory of Planned Behavior (TPB). This theory suggests that social factors, such as recommendations from friends, family, and social networks,

heavily impact purchasing decisions and consumer loyalty. When customers encounter positive reviews on social media, their likelihood of remaining loyal to a brand increases, and they may even become brand advocates [25]. This aligns with the Satisfaction-Loyalty Model, which suggests that satisfied customers are more likely to make repeat purchases. Individual characteristics, such as age, occupation, and education, play a crucial role in purchasing behavior. Younger individuals tend to prefer online shopping, whereas older customers are more inclined toward traditional shopping methods. Additionally, occupation and income influence consumers' preferences for luxury or discounted brands. Higher education levels are also associated with more thorough product comparisons and brand evaluations [24]. Numerous studies have explored the impact of economic conditions on customer satisfaction and, consequently, loyalty. For example, research studies [26,27] suggest that socio-economic status has a significant positive effect on customer satisfaction. Based on Oliver's (1999) Loyalty-Satisfaction Theory, satisfaction, in turn, positively influences loyalty [23]. Thus, lower-income customers tend to be more price-sensitive and responsive to discounts, while higher-income customers exhibit greater brand loyalty to companies that align with their values [28]. Moreover, behavioral analysis models such as RFM (Recency, Frequency, Monetary Value) which focus on how recently a customer has purchased, how often they buy, and how much they spend can help identify loyal customers and optimize marketing strategies [29]. Given the literature review, it is evident that economic, social, cultural, and emotional factors play a significant role in customer loyalty. Many comprehensive studies have addressed these aspects, including research [8], which provides further insights into these dynamics.

Most definitions of customer loyalty focus on the affection and level of commitment toward a product, service, brand, or organization, reflecting customers' intentions for repeat purchases. Thus, loyalty represents a customer's emotional, psychological, and subconscious need to receive a consistent source of value and satisfaction. One of the most common behaviors among loyal customers includes maintaining the relationship, expanding the scope of engagement, providing positive recommendations, engaging in word-of-mouth (WOM) marketing, and repurchasing the company's products [30,31]. According to research [30], customer loyalty is reflected in various behaviors, such as the number of purchases, time spent on a website or in-store, and the frequency of visits to both. Given this, analyzing customer shopping baskets is essential to understanding customers' shopping lists, purchase frequency, and repurchase behavior over time. Additionally, basket analysis can be leveraged through recommendation systems to offer personalized suggestions that enhance customer loyalty. Understanding customer behavior dynamics and analyzing hidden behavioral patterns is also crucial. These insights help identify unexpected shifts in purchasing behavior, which are essential for predicting long-term loyalty and developing effective customer retention strategies.

Customer Lifetime Value (CLV) is a key metric that businesses use to assess the long-term profitability of their customer base. CLV represents the financial value a customer generates throughout their relationship with a business. Analyzing CLV provides critical insights into how customer experience strategies impact long-term behavior and loyalty [32]. Several factors are considered when calculating CLV, including Average Order Value (AOV), Purchase Frequency, and Customer Churn Rate [32]. Given these factors, understanding customer behavior and emotions, as well as cultural, social, and economic conditions, plays a crucial role in companies' decision-making processes to enhance customer loyalty. In this regard, machine learning techniques can help simulate customer purchasing behavior and predict loyalty trends. These algorithms identify customer patterns and contribute to shaping effective marketing strategies [29]. In the relentless pursuit of competitiveness and profitability, businesses in the e-commerce sector have leveraged the power of advanced studies and technologies, particularly Machine Learning (ML), to optimize their decision-making processes. This optimization is not just an exercise in efficiency; it has become a catalyst for precise advertising and marketing strategies. In this dynamic e-commerce ecosystem, where traditional physical business boundaries no longer apply, businesses have evolved through a series of studies and research, uncovering new pathways for growth and efficiency.

Among these advancements, Machine Learning has emerged as a transformative force. Machine learning algorithms, which utilize vast amounts of data, offer an exciting promise: the ability to predict and adapt to customer demand in real-time, fundamentally transforming decision-making processes [9]. In today's competitive world, organizations store all their

customer information, and every organization, especially multichannel retailers, has access to vast amounts of data about their customers. Therefore, the importance of using multidimensional approaches that not only help accurately predict customer loyalty but also provide practical solutions for business growth and sustainability has become increasingly evident.

The theoretical framework serves as the intellectual and conceptual foundation of a research study, guiding the research process. This framework includes theories, models, and concepts upon which the study is designed and conducted. In the present study, the theoretical framework helps in better understanding key variables, their relationships, and predicting outcomes. Additionally, it provides a basis for formulating research hypotheses and allows for a comparison of the study's findings with previous research. In this study, by referring to relevant theories, an effort has been made to establish a solid foundation for examining the subject and ensuring a logical connection between theoretical foundations and research objectives. Some of the theoretical frameworks of loyalty are presented in Table 1.

The proposed model of this study aims to predict customer loyalty in a multichannel retail environment using machine learning techniques such as GCN, Transformer. In this regard, RFM and CLV are considered as key variables in analyzing customer purchasing behavior. Hofstede's cultural dimensions are incorporated as an influencing factor on customer behavior, and the Theory of Planned Behavior is applied through sentiment analysis of customers and its impact on future purchases. Data analysis revealed that combining social, economic, and cultural data with machine learning methods improves customer loyalty prediction. For example, customers with higher scores in the RFM model showed a greater tendency towards loyalty. Customers with a high CLV were more likely to repeat purchases. Cultural influences (such as collectivism and uncertainty avoidance) played an important role in customer preferences and their level of loyalty. Therefore, the theoretical framework proposed in this study establishes a clear connection between accepted theories in the field of customer loyalty and the analytical methods used.

Despite numerous studies on customer loyalty in multi-channel retail environments, significant gaps still exist in this area. Most previous research has focused on analyzing sales data and online customer behavior, addressing only limited factors such as purchase history and transaction data. However, a more accurate understanding of customer behavior requires analyzing multi-channel data and examining social, economic, and cultural variables that significantly influence purchasing decisions and customer loyalty. Additionally, many studies have been limited to transactional data and online customer behavior, with the impact of factors such as economic and cultural conditions on customer loyalty not being thoroughly investigated. Advanced machine learning models, such as Graph Convolutional Networks (GCNs) and Transformer models, can simulate complex and nonlinear relationships in data and provide more accurate predictions. However, the use of these models for predicting customer loyalty has not yet been extensively explored. Many existing studies on loyalty have focused primarily on loyalty drivers or predicting loyalty using sales data and online behavior analysis, often concentrating on one or two sales channels (such as physical stores or online platforms). In contrast, customer interactions with retailers occur across multiple channels, including websites, physical stores, and social media platforms. To address these research gaps, this study aims to develop and expand loyalty prediction models by considering not only transactional data but also social, cultural, economic, and emotional factors, RFM analysis, Customer Lifetime Value (CLV), churn rate, and customer behavior analysis—including shopping cart analysis, customer behavior dynamics, latent behavioral patterns, and sales data—to achieve more accurate predictions of loyalty.

While prior studies have explored customer loyalty using transactional or behavioral data, few have integrated socio-cultural, emotional, and macroeconomic variables into a unified predictive framework. Moreover, traditional models often neglect the dynamic nature of loyalty in omnichannel environments.From an empirical perspective, most previous research has relied on either survey-based data or limited-channel transactional records, often lacking large-scale, real-world data that includes cross-channel behaviors. In contrast, our study uses over 300,000 real omnichannel transactions from a multinational retailer, capturing both online and offline behavior across diverse geographic, cultural, and economic contexts.Furthermore, prior works rarely conduct loyalty prediction at both the individual and neighborhood

**Table 1. Key theories in customer loyalty.**

| Theories | Description |
|---|---|
| RFM Model | This model is based on three key criteria: Recency, Frequency, and Monetary (RFM). It is one of the primary methods used in customer loyalty analysis. In the present study, it has been utilized as a significant indicator for predicting customer loyalty. |
| CLV Theory | The CLV theory explains the total value a customer brings to a business over time. In this study, CLV is considered a key variable in predicting customer loyalty, as customers with a high CLV tend to show greater loyalty. |
| HCD Theory | Hofstede introduced five cultural dimensions to analyze consumer behavior. These dimensions include individualism vs. collectivism (where social recommendations influence purchasing behavior more in collectivist cultures) and uncertainty avoidance (societies with high uncertainty avoidance tend to show greater loyalty to well-known brands). In this study, cultural variables have been incorporated alongside other factors to examine the cultural effects on customer loyalty. |
| TPB | This theory suggests that a customer's behavioral intention is influenced by attitude, social norms, and perceived control. In this study, sentiment analysis of customer opinions using machine learning models has been employed as a representation of their attitude and purchase intent. |
| Consumer Behavior Model | The Howard-Sheth theory introduces a model of consumer behavior that examines various factors (such as motivations, selection criteria, and available information) that influence consumer purchase decisions. This model particularly explains the impact of social, economic, and cultural factors on buying behavior and customer loyalty [33]. |
| Economic conditions | It analyzes consumer behavior and shows that economic conditions, such as income and inflation, play a significant role in purchasing decisions. This book explains that customers' purchasing power is affected by the economic situation, and this can lead to changes in loyalty and brand preferences [34]. |

HCD: Hofstede's Cultural Dimensions Theory.

TPB: Theory of Planned Behavior.

levels simultaneously. Our dual-level modeling framework not only improves prediction accuracy but also enables practical CRM strategies tailored by customer segment and region, which is seldom addressed in existing empirical studies.This research addresses both theoretical and empirical gaps by combining sentiment analysis (via BERT), cultural dimensions (Hofstede), and economic indicators with advanced learning models such as GCN and Transformer. Unlike earlier studies that rely solely on RFM or CLV, our hybrid approach enhances prediction accuracy by modeling complex behaviors within a dynamic multichannel retail environment.

These variables can have a profound impact on customer purchase decisions, making it essential to simulate customer behavior more effectively by considering these factors. In addition to traditional models, this study employs advanced machine learning techniques, such as Graph Convolutional Networks (GCNs) and Transformer models, to analyze complex datasets and provide more precise loyalty predictions. These algorithms can simulate customer behavior with greater accuracy and offer optimized solutions for marketing strategies. To bridge the research gaps, an initial baseline model has been implemented to predict customer loyalty in a simplified manner. This baseline model excludes social, economic, and cultural variables, serving as a foundation for further model enhancements and refinements. This baseline model serves as a reference point for comparison with more advanced models that incorporate these additional factors. By combining these models, we can evaluate the impact of these factors on loyalty prediction accuracy and demonstrate how integrating

them enhances prediction precision. In many cases, complex relationships exist within the stored organizational data, necessitating robust approaches to identify and analyze these intricate connections. While the use of machine learning techniques in predicting customer behavior and loyalty is increasing, this research gap presents new opportunities for applying advanced machine learning techniques such as Graph Convolutional Networks (GCNs), Transformer models. These techniques enable the analysis of complex datasets and more accurate customer loyalty predictions.

Additionally, scenario analysis using reinforcement learning (Q-learning) helps simulate various economic, social, and cultural conditions and assess their impact on customer loyalty. Therefore, the key research questions of this study are as follows:

1. How can multi-channel data and social, economic, and cultural variables be utilized to more accurately predict customer loyalty in multi-channel retail?

2. How can the use of advanced machine learning models, such as Graph Convolutional Networks (GCNs) and Transformer models, improve the accuracy of customer loyalty prediction compared to traditional models?

The main objective of this research is to develop a model that simultaneously combines multichannel data with social, economic, and cultural factors to predict customer loyalty in a multichannel retail environment. This model employs advanced machine learning methods such as Graph Convolutional Networks (GCN) and Transformer models, offering higher accuracy in predicting customer loyalty compared to traditional models like RFM and CLV, which focus solely on transactional data. The remainder of this study is structured as follows: Section 2 is dedicated to reviewing the literature and related studies. Section 3 explains the methods and problem-solving approaches. Section 4 describes the implementation process, while Section 5 analyzes the results. In Section 6, the model is compared with the baseline model. In Section 7, sensitivity analysis is conducted through scenario modeling. Section 8 presents the practical implementation of the loyalty prediction model within CRM/EIS systems, demonstrating how it can support real-world operational decisions through integration with platforms like SAP, Oracle, and Salesforce. Finally, the last section presents discussion and recommendations for future research.

## 2. Literature review

Consumer loyalty is considered the most desirable phenomenon for companies, especially in the omnichannel space, because commitment leads to repeat purchase behavior and positive word-of-mouth advertising. Creating efficient prediction models for loyalty is one of the important concerns of the marketing manager. Accurate prediction is a challenging task due to the statistical complexities caused by nonlinear consumer behavior. Research has shown that customer loyalty is influenced by factors such as shopping experience, service quality, and even social, cultural, and economic factors. Given that omnichannel retailing has led to multichannel customer interaction, understanding and predicting customer loyalty effectively requires advanced techniques such as machine learning. Hence, recent research has focused on using learning models to analyze customer data, but most of these studies have little focus on cultural, social, and economic data. Some researchers have investigated the important and influential factors on loyalty. Indicators affecting loyalty are very important because by identifying these factors, a big step can be taken towards customer satisfaction and loyalty. For further study, you can refer to the research of [35–41]. As mentioned in the previous section, loyalty affects important decisions of an organization because attracting new customers will cost an organization more than retaining existing customers. On the other hand, not all customers are equally useful to the organization, and organizations must identify their effective customers, or, in other words, their loyal customers. Therefore, many researchers have studied the prediction of customer behavior. As far as we know, the only research that has studied the prediction of loyalty in the field of omnichannel retail is the research of [42]. He and his colleagues first identified the different parts of the customer journey and then examined the strength of the customer journey segments with the advent of mobile devices and finally addressed which components of the customer experience (i.e., product satisfaction, journey satisfaction, and customer inspiration) have

the greatest contribution to customer loyalty for distinct customer journey segments. Finally, using a combined relationship between the aforementioned components, they presented a mathematical relationship for customer loyalty. A notable point in this study is that they only predicted loyalty by surveying customers. However, it is noteworthy that customers will not always respond, and access to customers is not easy, so the need to predict customer loyalty through data recorded by the organization becomes important, which has not been done by researchers so far and is the only one that this study has addressed. Some of the researchers' research in the field of loyalty prediction is presented in Table 2.

According to the literature review, numerous studies exist in the field of predicting and analyzing customer purchasing behavior; however, many of these studies have focused more on examining factors that influence loyalty or predicting factors such as customer churn, satisfaction, and so on. As shown in the table, in most previous studies, omnichannel interactions, which have become increasingly common in today's retail landscape, have been overlooked, and analyses have been influenced by transactional data and specific channels (such as online shopping or physical stores). While several studies, including research [75] that have predicted customer behavior using advanced learning models, emphasize the importance of using omnichannel data in predicting customer behaviors and have stated that the use of omnichannel data can significantly enhance predictive power, fewer studies have examined the impact of factors such as cultural, social, and economic conditions, emotions, and customer behavioral traits alongside traditional CLV and RFM analyses on loyalty behavior in a multichannel environment.

While these factors can play a crucial role in shaping customer behavior, many loyalty prediction models have been limited to analyzing purchase data or customer interactions with brands. Therefore, the gap in the literature has prevented analyses from offering a more comprehensive picture of customer loyalty behavior in multichannel retailing. Given the importance of loyalty and the rise of omnichannel retailers, developing a model to predict loyalty using the mentioned characteristics is of great significance. In practice, loyalty prediction is also highly significant. Major companies such as Amazon and Walmart have leveraged loyalty prediction models to enhance customer experience and increase retention [84]. By analyzing comprehensive data from various channels (including online stores, mobile applications, and in-store interactions), these brands have been able to simulate customer behavior and predict loyalty. The use of these models has enabled them to make more informed decisions regarding marketing strategies and customer experience optimization. Given the limitations of previous research, future studies could focus on integrating data from all channels (both online and offline) to develop more accurate loyalty prediction models. Additionally, considering social, cultural, and economic factors alongside other behavioral characteristics of customers could significantly improve loyalty predictions and provide a deeper understanding of customer behavior. In summary, it can be said that many previous studies on customer loyalty prediction models have primarily focused on the use of transactional data and customer behavior analysis models, such as RFM and CLV, which are less effective in multichannel environments. While these models have been able to predict customer loyalty to some extent, the lack of social, economic, and cultural data in these analyses reduces the accuracy of predictions in more complex conditions. This study aims to fill this gap by using multichannel data and combining it with social and economic factors to develop a model that can provide more accurate predictions of customer loyalty.

## 3. Materials and methods

The main objective of this study was to predict customer loyalty of an omnichannel retailer by including variables such as omnichannel data, customer demographic characteristics, customer sentiment towards products and the retailer, purchasing behavior, the impact of cultural, social, and economic conditions on purchasing power, customer lifetime value (CLV), and RFM (Customer Lifetime Value (CLV) and RFM have been used to construct the loyalty variable and are not considered in the prediction of loyalty). To achieve this goal, two main approaches have been used. First, customers are divided into neighborhoods based on different characteristics, and then the loyalty probability for each neighborhood and each customer is predicted separately using learning techniques. Predicting loyalty at both the neighborhood and individual customer levels is crucial because it significantly influences managerial decisions such as selecting products to be

**Table 2. Literature review.**

| Research | Channel Type: Omni-channel retail | Channel Type: Other | Objective | Variables Under Study: Other Data | Omni-channel Data | RFM | Social Conditions | Impact of Economic Conditions | Impact of Cultural Conditions | Customer Lifetime Value | Hidden Customer Behavior Patterns | Customer Behavior Dynamics | Customer Sentiments | Customers' Shopping Cart | Customer Churn Rate | Customer Segmentation | Solution Method |
|---|---|---|---|---|---|---|---|---|---|---|---|---|---|---|---|---|---|
| [43] | * | | Loyalty/brand | * | | | | | | | | | | | | | Dirichlet |
| [44] | * | | Purchase | | * | | | | | | | | | | | | ML |
| [45] | * | | Purchase | | * | | | | | | | | | | | * | ML |
| [46] | * | | Loyalty Analysis | | * | | | | | | | | | | | | SEM |
| [47] | * | | Loyalty Analysis | | * | | | | | * | | * | | * | | * | CRM Methods and Traditional Data Analysis |
| [14] | | * | Prediction of Loyalty | | | * | | | | | | | | | | | ML |
| [48] | * | | Loyalty/brand | | | | * | | * | | | | | | | | Econometric Model |
| [49] | * | | Loyalty/brand | | * | | | | | | | * | | * | | | Empirical Research and Statistical Analysis |
| [50] | * | | Loyalty Analysis | | * | | | | | | | | | | | | SEM |
| [51] | * | | Loyalty Analysis | | * | | | | | | | * | | * | | | Analytical and Mixed Methods |
| [52] | | * | Purchase | | | * | | | | | * | * | | * | | * | Data Mining |
| [53] | | * | Purchase | * | | * | | | | | * | * | | | | * | Data Mining |
| [54] | | * | Loyalty | | | | | | | | * | * | | * | | * | ML |
| [55] | * | | Customer Segmentation | | * | | | | | | | | | | | | LCCA |
| [56] | * | | Loyalty Analysis | * | | | | | | | | | | | | | Qualitative Research |
| [57] | * | | Customer Journey | | | | | | | | | | | | | | customer journey maps |
| [58] | | * | Loyalty Analysis | * | | | | | | | * | | | | | * | Data Mining |
| [59] | | * | Loyalty Analysis | | | * | | | | | | | | | | * | ML |

*(Continued)*

Table 2. (Continued)

| Research | Channel Type | | Objective | Variables Under Study | | | | | | | | | | | | | Solution Method |
| --- | --- | --- | --- | --- | --- | --- | --- | --- | --- | --- | --- | --- | --- | --- | --- | --- | --- |
| | Omni-channel retail | Other | | Other Data | Omni-channel Data | RFM | Social Conditions | Impact of Economic Conditions | Impact of Cultural Conditions | Customer Lifetime Value | Hidden Customer Behavior Patterns | Customer Behavior Dynamics | Customer Sentiments | Customers' Shopping Cart | Customer Churn Rate | Customer Segmentation | |
| [60] | * | | Loyalty Analysis | | * | | | | | | | | | | | | Analytical Method |
| [61] | * | | Loyalty Analysis | | * | | | | | | | * | | | | | Consumer Behavior Analysis |
| [62] | * | | Loyalty Analysis | | * | | * | | | | | | | | | | Survey |
| [63] | * | | Loyalty Analysis | | * | | | | | | * | | | | | | Conceptual analysis |
| [64] | * | | Loyalty Analysis | | * | * | * | | * | | | | | | | * | SEM |
| [65] | | * | Churn | * | | | | | | | | | | | | * | Data Mining & ML |
| [66] | | * | Churn | * | | | | | | | | | | | | | ML |
| [37] | | * | Loyalty Analysis | * | | | | | | | | | | | | | ML |
| [67] | | * | Loyalty Analysis | * | | | * | * | | | | | | | | * | Data Mining |
| [68] | | * | Churn | * | | | | | | | * | | | | | * | Data Mining & ML |
| [39] | | * | Loyalty Analysis | * | | | | | | | | | | | | * | Data Mining & M |
| [69] | | * | Loyalty Analysis | * | | | | | | | | | | | | * | ML |
| [70] | | * | Churn | * | | | | | | | | | | | | | ML |
| [71] | | * | Churn | * | | | | | | | | | | | | * | ML |
| [72] | | * | Churn | * | | | | | | | * | | | | | * | DL & ML |
| [16] | * | | Loyalty Analysis | * | | * | | | | | | | | | | | |
| [73] | * | | Loyalty Analysis | | * | | | | | | | | * | | | | Statistical Analysis |
| [74] | * | | Loyalty Analysis | | * | | | | | | | * | | | | | Flow Theory and Hyperbolic Discounting Theory |
| [75] | * | | Loyalty Analysis | * | * | | | | | | | | | | | | SEM and CFA |
| [76] | * | | Loyalty Analysis | | * | | | | | | | | | | | | PLS-SEM |

(Continued)

Table 2. (Continued)

| Research | Channel Type | | Objective | Variables Under Study | | | | | | | | | | | | | Solution Method |
|---|---|---|---|---|---|---|---|---|---|---|---|---|---|---|---|---|---|
| | Omni-channel retail | Other | | Other Data | Omni-channel Data | RFM | Social Conditions | Impact of Economic Conditions | Impact of Cultural Conditions | Customer Lifetime Value | Hidden Customer Behavior Patterns | Customer Behavior Dynamics | Customer Sentiments | Customers' Shopping Cart | Customer Churn Rate | Customer Segmentation | |
| [35] | * | | Purchase | | * | | | | | | | * | | | | | ML |
| [77] | * | | Purchase | | * | | | | | | | | | | | | DL |
| [78] | * | | Loyalty Analysis | | * | | * | | | | | | | | | | Empirical Measurement and Psychometric Testing Methods |
| [79] | * | | Loyalty Analysis | | * | | | | | | | | | | | | Data-driven Research |
| [41] | * | | Loyalty Analysis | | * | | | | | | | | | | | | PLS-SEM |
| [80] | * | | Loyalty Analysis | | * | | * | | | | | | | | | | Deep Interviews |
| [81] | * | | Loyalty Analysis | | * | | | | | | | | | | | * | ML |
| [40] | * | | Loyalty Analysis | | * | | | | | | | | | | | | Survey |
| [82] | * | | Loyalty Analysis | | * | | | | | | | | | | | * | SEM |
| [83] | * | | Purchase | | * | * | | | | | * | * | | | | | DL & ML |
| [84] | | * | Loyalty Analysis | * | * | | | | | | * | * | | | | * | ML |
| [85] | * | | Loyalty Analysis | * | * | | | | | | | | | | | | PLS-SEM |
| [86] | * | | Loyalty Analysis | | * | | | | | | | | | | | | Topic Modeling |
| [87] | * | | Loyalty Analysis | | * | | | | | | | | | | | * | Self-Determination Theory |
| [88] | | * | Loyalty Analysis | * | | | | | | | | | | | | | Mixed-Methods |
| [89] | | * | Customer Satisfaction | | | | | | | | | | | | | | SEM |
| [90] | | * | Purchase Behavior | * | | | | | | | | | | | | | PLS-SEM |
| [91] | | * | Loyalty Analysis | * | | | | | | | | | | | | | PLS-SEM |

(Continued)

Table 2. (Continued)

| Research | Channel Type | | Objective | Variables Under Study | | | | | | | | | | | | | Solution Method |
|---|---|---|---|---|---|---|---|---|---|---|---|---|---|---|---|---|---|
| | Omni-channel retail | Other | | Other Data | Omni-channel Data | RFM | Social Conditions | Impact of Economic Conditions | Impact of Cultural Conditions | Customer Lifetime Value | Hidden Customer Behavior Patterns | Customer Behavior Dynamics | Customer Sentiments | Customers' Shopping Cart | Customer Churn Rate | Customer Segmentation | |
| [92] | | * | Loyalty Analysis | * | | | | | | | | | | | | | Regression Analysis |
| [93] | | * | Prediction of Loyalty | * | | | | | | | | | | | | | PLS-SEM |
| [94] | | * | Customer Churn | * | | | | | | | | | | | | | ML |
| [95] | | * | Customer Churn | * | | | | | | | | | | | | | ML |
| [96] | | * | Prediction of Loyalty | * | | | | | | | | | * | | | | ML |
| [97] | | * | Loyalty Analysis | * | | | | | | | | | | | | | SEM |
| [98] | | * | Purchase Behavior | * | | | | | * | | | | | | | | Multilevel Modeling |
| [99] | | * | Loyalty Analysis | * | | | | * | | | | | | | | | PLS-SEM |
| [100] | | * | Loyalty Analysis | * | | | | * | | | | | | | | | Bibliometric Methods |
| [101] | | * | Loyalty Analysis | * | | | | | | | | | | | | | Bayesian model |
| [102] | | * | Customer Behavior | * | | | | | | | | | | | | | ML |
| [103] | | * | product choice prediction | * | | | | | | | | | | | | | ML |
| [104] | | * | shaping consumer behavior | * | | | | | | | | | * | | | | Panel Data Regression Analysis |
| [105] | | * | Customer satisfaction | * | | | | | | | | | * | | | | ML |
| [106] | | * | Prediction of Loyalty | * | | | | | | | | | * | | | | PLS |

(Continued)

Table 2. (Continued)

| Research | Channel Type | | Objective | Variables Under Study | | | | | | | | | | | | | | Solution Method |
|---|---|---|---|---|---|---|---|---|---|---|---|---|---|---|---|---|---|---|
| | Omni-channel retail | Other | | Other Data | Omni-channel Data | RFM | Social Conditions | Impact of Economic Conditions | Impact of Cultural Conditions | Customer Lifetime Value | Hidden Customer Behavior Patterns | Customer Behavior Dynamics | Customer Sentiments | Customers' Shopping Cart | Customer Churn Rate | Customer Segmentation | |
| [107] | | * | Customer Lifetime Value Prediction | * | | | | | | | | | | | | * | ML,DL |
| Current research | * | | Prediction of Loyalty | | * | * | * | * | * | * | * | * | * | * | * | * | DL and ML |

The term "Loyalty Analysis" means that the paper does not directly predict loyalty, but rather examines the factors that influence it. In other words, the focus of the paper is on identifying and analyzing the factors that affect loyalty, rather than making direct predictions about it.

PLS-SEM (Partial Least Squares Structural Equation Modeling).

Confirmatory Factor Analysis (CFA) methods.

Latent Class Cluster Analysis.

shipped to different areas, setting regional pricing strategies, and making other operational decisions. Therefore, customers were segmented into neighborhoods to more accurately assess loyalty both at a micro (customer) and macro (neighborhood) level. The approaches presented in Table 3 have been used to evaluate the model's efficiency.

In this study, advanced machine learning methods such as Graph Convolutional Networks (GCN) and Transformer models are used to predict customer loyalty in a multichannel retail environment. In addition to transactional data, the model utilizes social, economic, and cultural variables to simulate different conditions and more accurately predict customer behavior. Compared to traditional models like RFM and CLV, which primarily focus on simple historical data, the proposed model performs better, especially under more complex economic and social conditions.

Customer loyalty plays a critical role in the long-term success of organizations, especially omnichannel retailers. Retailers, who typically have physical stores in various locations, face a complex challenge in understanding and predicting customer loyalty. Customer loyalty is not a static concept; it is influenced by factors such as customer preferences, external economic conditions, and even social and cultural influences, all of which change over time. Therefore, accurately predicting customer loyalty is not solely possible by analyzing past transactions; it requires examining dynamic behaviors and external factors that affect purchase decisions. Predicting customer loyalty, especially in a market where customer behaviors are constantly changing, is challenging. In addition to fluctuations in customer preferences, external economic conditions, social influences, and even cultural changes also play a significant role in these shifts. As a result, to effectively predict customer loyalty, a model must be developed that can adapt to these complexities. These challenges include changes in customer behavior and the impact of various environmental factors on their purchase decisions. Given the inclusion of multidimensional variables and the dynamic nature of this topic due to shifts in customer preferences and tastes, the complexity of the problem has increased. Therefore, a model must be employed that provides accurate predictions of customer loyalty. Recent research in the field of predicting customer loyalty and behavior has shown that machine learning algorithms are highly effective in predicting customer purchasing behaviors. These studies indicate that the use of advanced machine learning methods can enhance prediction accuracy and help better understand customer behavior. An example of this research is presented in Table 4.

This study considers two main approaches for predicting customer loyalty: individual-level prediction and neighborhood-level prediction. These two approaches are particularly important for Omni-channel retailers who have physical stores in different regions. Retailers need to adjust their strategies—such as shipping quantities, pricing, and discounts—based on the specific needs of each region. It is also important to note that not all customers are equally valuable to organizations. Therefore, understanding the likelihood of loyalty for each customer in each neighborhood can help retailers make better decisions. Additionally, since Omni-channel retailers possess vast amounts of data, accurately predicting customer loyalty becomes particularly crucial. Incorrect loyalty predictions can lead to significant losses, making the selection of an appropriate prediction model critically important. Besides transaction data, variables such as economic

**Table 3. Prediction models.**

| Model | Learning Approaches | | | Model Strategy | | Prediction Type | |
|---|---|---|---|---|---|---|---|
| | Suprvise | Unsupervise | Reinforcement | Hybrid Approach | Non-Hybrid Approach | Loyalty for Customers | Loyalty for Neighbors |
| Autoencoder &classification model | ∗ | ∗ | | ∗ | | ∗ | ∗ |
| PCA & Random Forest | ∗ | ∗ | | ∗ | | ∗ | ∗ |
| XGBoost & LSTM | ∗ | ∗ | | ∗ | | ∗ | ∗ |
| Particle Swarm Optimization & LSTM | ∗ | | | ∗ | | | ∗ |
| CNN | ∗ | | | | ∗ | ∗ | ∗ |
| Adaptive GCN-Transformer Hybrid | ∗ | | | ∗ | | ∗ | ∗ |

**Table 4. Some studies related to customer behaviors using learning techniques.**

| Study | Objective | Algorithm |
|---|---|---|
| [77] | Customer Journey | Deep Learning Algorithm |
| [29] | Customer Behavior | Random Forest, Artificial Neural Network, Support Vector Machine, K-Nearest Neighbors, Naive Bayes, Logistic Regression, AdaBoost, XgBoost, Stochastic Gradient Descent, and Hybrid Algorithms |
| [53] | Customer Behavior | Logical Decomposition Analysis and Decision Tree |
| [108] | Customer Churn | Artificial Neural Network, Support Vector Machine, Decision Tree, Naive Bayes, and Logistic Regression Algorithms with their Enhanced Versions |
| [72] | Customer Churn | Logistic Regression, Naive Bayes, Random Forest, Decision Tree, K-Nearest Neighbors, and ANN |
| [69] | Repeat Purchase | Random Forest, Gradient Boosting, XgBoost |
| [58] | Loyalty | C4.5, Naive Bayes, and Nearest Neighbor |
| [16] | Loyalty | Gradient Boosted Decision Tree Classifier – Best for Binary and Random Forest Classification |
| [84] | Loyalty Evaluation | Semi-Supervised Approach |
| [109] | Customer Churn | Fully Connected Layer Convolutional Neural Network – Long Short-Term Memory (FCLCNN-LSTM) |
| [110] | Customer Churn | LSTM |
| [111] | Understanding Human Actions | Multi-scale Graph Convolutional Network (MGCN) |

conditions, social influences, cultural factors, customers' shopping baskets, customer lifetime value, and hidden behavioral patterns are also considered in this research. These features enhance the model's prediction power and adaptability.

Various studies have used machine learning techniques to predict factors related to customers, such as purchase likelihood, satisfaction, churn rate, and loyalty. For instance, studies referenced in [88–96] confirm the use of advanced methods such as XGBoost and LSTM for analyzing complex, multi-dimensional data. These models can capture complex relationships between various variables (such as buying behavior, social, economic, and cultural factors), making them particularly suitable for predicting customer loyalty. XGBoost and LSTM are particularly well-suited for processing large, complex datasets. The tree-based XGBoost approach can effectively model non-linear relationships in data, while LSTM is excellent for sequential or time-series data (such as customer purchase history). Since customer loyalty constantly changes and depends on customer preferences, LSTM can model time-dependent patterns in customer behavior and provide more accurate predictions. By combining XGBoost and LSTM in a hybrid model, the benefits of both approaches are harnessed, enhancing the accuracy of loyalty predictions. This hybrid approach has the advantage of processing different features of the data: XGBoost handles non-temporal features, while LSTM handles time-dependent ones.

An innovative approach employed in this study is the adaptive GCN-Transformer hybrid model integrated with a recommendation system. This hybrid approach combines Graph Convolutional Networks (GCN) and Transformers. GCN is used to analyze complex relationships between customers and products, while the Transformer processes time-series data to simulate changes in customer preferences over time. By merging these two models, the system can dynamically update its predictions and model customer behavior based on changes in preferences and their responses to various recommendations. The GCN model is especially effective for identifying hidden relationships between customers based on interactions, purchases, or social connections. This model can detect patterns in customer data that may not be visible through traditional methods, contributing to more accurate segmentation and better predictions. On the other hand, the Transformer model is excellent for processing time-series data such as customer purchase history, which is crucial for understanding temporal changes in customer behavior. Integrating these models into a recommendation system significantly enhances prediction accuracy and the personalization of recommendations. Traditional recommendation systems

typically focus on collaborative filtering or content-based filtering for suggesting products. However, incorporating the GCN model into the recommendation system allows it to consider more complex relationships between customers and products, resulting in more accurate recommendations. Additionally, since customer preferences evolve over time, the Transformer model ensures that the recommendation system can dynamically simulate these changes. By leveraging sentiment analysis, the system can simulate customer sentiments towards various products and improve the quality of recommendations. This feature, less emphasized in traditional recommendation systems, helps to make suggestions more accurate and personalized based on customers' emotional responses.

The combination of graph-based models and time-series predictions provides a deeper understanding of customer behavior, which is essential for making more informed decisions about customer loyalty and personalized marketing strategies. In other words, the proposed combined GCN and Transformer model can effectively capture intricate patterns and dynamic interactions within the data, improving the accuracy of predictions while remaining flexible enough to adjust to different market and environmental changes. Table 5 describes the loyalty prediction methods along with a brief explanation of their approach.

In recent years, sentiment analysis has been enriched by deep learning techniques, particularly transformer-based models such as BERT (Bidirectional Encoder Representations from Transformers). BERT has demonstrated advanced performance in text classification tasks, including customer sentiment analysis in retail contexts [112–114]. The selection of BERT, GCN, Transformer, and Reinforcement Learning (Q-Learning) models in this study was driven by the need to address the multidimensional and complex nature of customer loyalty within a multi-channel retail environment. Unlike

**Table 5. Model approaches.**

| Model | Brief explanation |
|---|---|
| Autoencoder & classification model | This model uses a hybrid approach combining supervised and unsupervised learning, incorporating an autoencoder and a classification model. Data is preprocessed, normalized, and one-hot encoded. The autoencoder reduces dimensionality and extracts encoded features, which are then used in a classification model to predict customer loyalty. |
| PCA & Random Forest | The code uses a hybrid approach, combining supervised and unsupervised learning methods, including PCA for dimensionality reduction and Random Forest for classification. Data is preprocessed, normalized, and encoded, then PCA is applied for dimensionality reduction. The model is trained, evaluated |
| XGBoost & LSTM | This code uses a hybrid approach with two models to predict customer loyalty: XGBoost: Used to predict customer loyalty scores based on various features. LSTM: Used for analyzing temporal patterns and improving predictions by incorporating the predictions from XGBoost as additional features. |
| Particle Swarm Optimization & LSTM | The PSO algorithm is used to optimize the hyperparameters of the LSTM model, such as the number of units in the LSTM layers and the dropout rate. After finding the best parameters, the LSTM model is trained with these parameters and evaluated to make more accurate predictions of customer loyalty. |
| CNN | The model uses a Convolutional Neural Network (CNN) model to predict customer loyalty based on numerical and categorical data. The model is preprocessed, trained, and evaluated using performance metrics. The output is converted into a probability between 0 and 1. |
| Adaptive GCN-Transformer Hybrid | This model creates a combined GCN-Transformer model to predict customer loyalty based on various features. The model uses two main components: GCN (Graph Convolutional Network) for processing the graph relationships of the data, and Transformer for analyzing temporal data. It also employs adaptive learning techniques for weighting and prediction, and a recommendation system has been added to provide personalized suggestions for users based on the available data. |

many purely behavioral phenomena, loyalty is a combination of emotional, social, economic, and behavioral factors that manifest across structured, unstructured, and network data. Traditional models such as Logistic Regression, RFM, or even XGBoost lack the capacity to effectively process this diversity.

BERT was chosen for its ability to analyze unstructured customer feedback text, extracting sentiments and understanding contextual and implicit language nuances with high accuracy. This capability surpasses older methods like TF-IDF or LSTM, which tend to be more superficial and lack deep contextual comprehension.

GCN (Graph Convolutional Network) was employed to model the network structure of customer interactions, such as participation in campaigns and shared reviews. GCN excels at identifying hidden network patterns and peer influences, which classical models usually ignore by treating customers as independent entities.

Transformer models were selected to handle sequential data, such as purchase histories and session logs, due to their superior performance in capturing long-term temporal and textual dependencies through self-attention mechanisms. Unlike LSTM models, which suffer from the vanishing gradient problem and memory loss over long sequences, Transformers maintain better contextual awareness.

Reinforcement Learning, specifically Q-Learning, was utilized to simulate marketing decision scenarios and customer purchase behaviors, enabling the modeling of long-term reward-based decision making. This approach overcomes the static nature of traditional predictive models by incorporating the dynamic interaction between decisions and loyalty outcomes.

By combining these models, the analysis extends beyond merely relying on past customer behaviors. It incorporates psychological dimensions via sentiment analysis with BERT, social relationships and network effects through GCN, and dynamic customer decision-making in response to campaigns through Q-Learning. This comprehensive approach addresses the key limitations of earlier studies that focused narrowly on RFM, CLV, or simple LSTM models.

Additionally, model design carefully considered the alignment between data type and model architecture. For instance, BERT was favored over LSTM/GRU for textual feedback due to its superior understanding of complex sentences, while Transformer was chosen for sequence modeling because of its more accurate self-attention mechanism compared to traditional RNNs.

From an implementation perspective, all models were trained on a combined loyalty index derived from SEM, and their outputs were integrated using a weighted ensemble approach that assigns weights based on each model's accuracy. This ensemble technique enhanced overall prediction accuracy and reduced forecast variability.

In summary, the choice of these architectures was guided not only by technical compatibility with data types but also by the specific needs of the problem and the aim to fill methodological gaps in the existing customer loyalty literature. So in this study, the selection of methods used to predict customer loyalty is based on a combination of theoretical and practical considerations. Graph Convolutional Networks (GCN) and Transformer models were chosen for their ability to model complex relationships between customers and analyze time-series data. This choice aligns with the theoretical framework of the research, as traditional models like RFM and CLV rely solely on transactional data and overlook social relationships and behavioral changes. In contrast, GCN enables the analysis of customer interactions within a graph-based space, while Transformer identifies temporal patterns in customer behavior. Furthermore, the selected statistical methods, such as XGBoost, LSTM, and Random Forest, were employed based on previous studies that demonstrated these algorithms' high efficiency in analyzing multi-dimensional customer data and predicting loyalty. Since customer loyalty is influenced by economic, social, cultural, and behavioral factors, combining these models allows the identification of complex patterns that cannot be detected with simple linear models. For this reason, the chosen machine learning models are directly related to the research questions, as they can simultaneously analyze interactions between customers, transactional data, and environmental factors. As mentioned earlier, a novel approach for predicting customer loyalty has been presented in this research. To understand the process of this approach, a summary of the model's implementation steps is provided in Algorithm 1 and Fig 1 and Table 6.

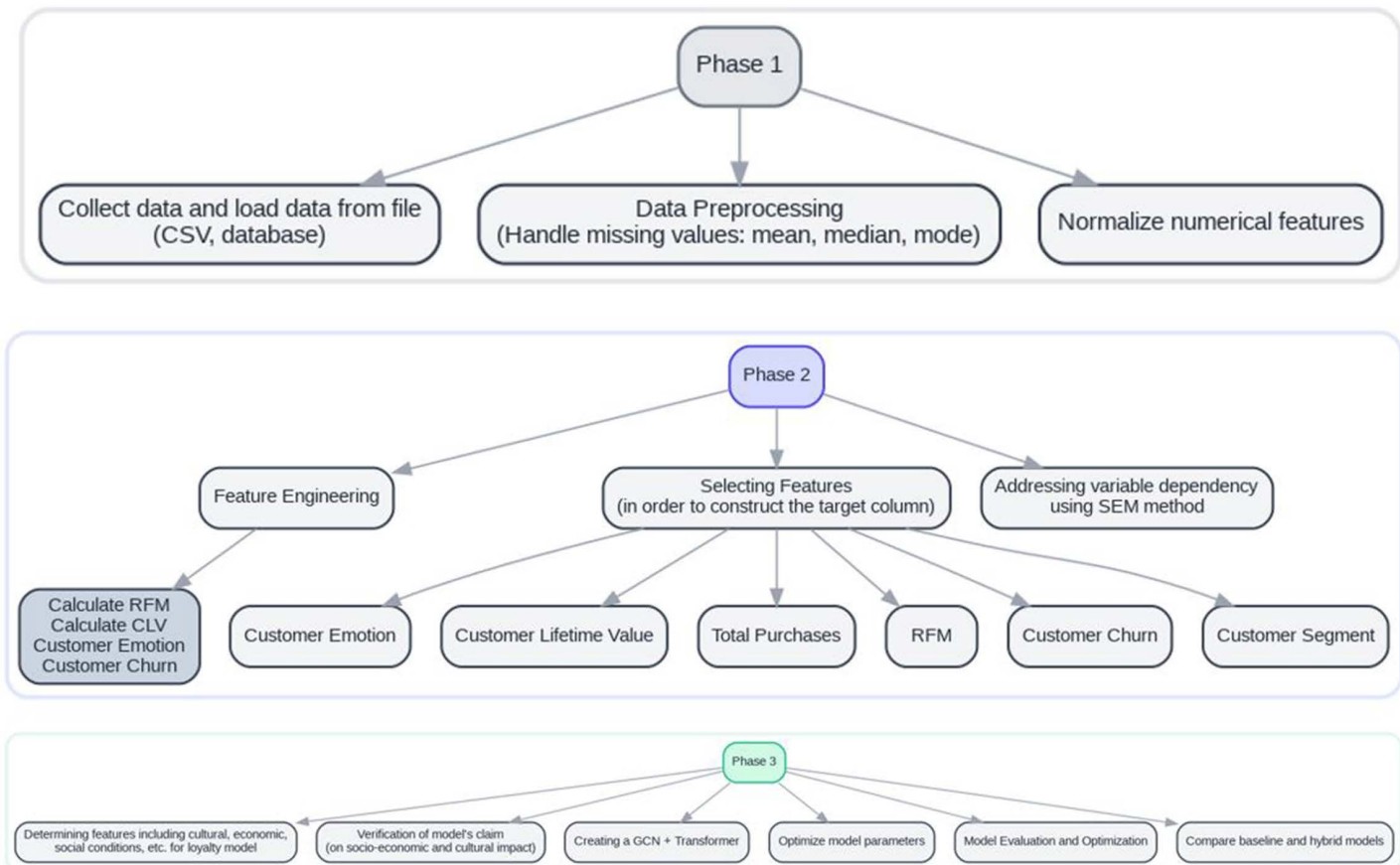

**Fig 1. Overall view of the steps performed.**

Algorithm 1. Theoretical framework and problem modeling.

| Step 1: Data Preparation | 1. Load data from file (CSV, database) | |
|---|---|---|
| | 2. Preprocess data: | Handle missing values (impute with mean/median/mode) |
| | | Normalize numerical features |
| | | Encode categorical features (One-hot/Ordinal) |
| | 3. Feature Engineering: | Calculate RFM (Recency, Frequency, Monetary) |
| | | Calculate CLV |
| | | Perform sentiment analysis using BERT and other featureS |
| Step 2: Feature Analysis and Model Selection | Define key features: RFM, CLV, Sentiment, Economic features, … | |
| | Identify patterns in customer behavior (Clustering) | |
| | Select models (e.g., Logistic Regression, XGBoost) | |
| Step 3: Feature Integration and Pre-processing for GCN-Transformer | Pre-predicted Features | Predicting the churn rate, customer basket, and customer sentiment in previous steps. |
| | | Feeding these predictions as new inputs to the GCN-Transformer model. |
| | Features Input to GCN-Transformer | RFM features (Recency, Frequency, Monetary), CLV, and economic and social features calculated in the previous step. |
| | | Pre-predicted features (churn rate, customer basket, sentiment) as input features. |
| | Preparation for GCN-Transformer Model: | Collecting and combining the predicted features into the GCN-Transformer model. |
| | | Preparing graph data for the next steps. |

| Step 4: Customer Relationship Analysis with Graph Neural Networks (GCN) | Build Customer-Product Interaction Graph | Nodes: Customers and Products |
| --- | --- | --- |
| | | Edges: Purchase interactions, Social influence, Similar behaviors |
| | Train GCN Model (Graph Convolutional Network) | Inputs: Node features and relationships (edges) between nodes |
| | | Train the Graph Convolutional Network to analyze complex relationships between customers and products |
| Step 5: Customer Loyalty Modeling with Adaptive GCN-Transformer Learning | Definition of the Adaptive Hybrid Model (GCN-Transformer) | Using the GCN model to analyze complex relationships between customers and products through graphs |
| | | Using Transformer to analyze time-series data and simulate changes in customer preferences |
| | | Combining the outputs of GCN and Transformer to learn nonlinear and hidden dependencies in the data. |
| | Adaptive Learning Feature | The model is continuously updated with new data and adapts to changes in customer behavior and market conditions |
| | | Using GCN to understand customer relationships through social interactions and shared purchases. |
| | | Using Transformer to analyze temporal changes and customer responses to different offers. |
| | Model Training | Inputs: Customer behavior data, purchase interactions, social influences, economic trends |
| | | Output: Prediction of customer loyalty score and simulation of customer responses to market changes. |
| Step 6: Designing a Dynamic and Personalized Recommendation System | Definition of the Hybrid Recommendation System | Content-Based Filtering: Recommending similar products based on product features |
| | | Collaborative Filtering: Recommending based on customers with similar behaviors. |
| | | GCN Model: For discovering hidden relationships between customers and products through networked relationships |
| | | Transformer Model: For analyzing temporal changes in customer preferences and their responses to ads and offers. |
| | Personalizing Recommendations Based on Loyalty and Customer Behavior | If customer loyalty is below threshold: Apply discount strategy or special offers |
| | | If the customer is loyal and active: Recommend popular and high-value products. |
| | | Consider social influences and shared customer behaviors for more accurate recommendations. |
| Step 7: Model Evaluation and Optimization | Evaluate Model Performance | Use evaluation metrics such as: |
| | Compare Baseline and Hybrid Models | Compare the results from different models and choose the best-performing one |
| | Optimize Model Parameters | Use techniques to tune hyperparameters |
| Step 8: Sensitivity Analysis and Scenario Simulation | Define Scenarios | Unemployment Rate, Inflation Rate, Customer Emotion, Cultural_impact, Social_impact |
| | Simulate Scenarios | Use Reinforcement Learning (Q-Learning) to predict customer responses |

**NOT:** The variables used in building loyalty are not included in the loyalty prediction model.

To manage the data, the following steps were performed:

**Missing Data Handling:** Median imputation was applied to numerical features (e.g., Age, Total Amount) due to its robustness against outliers. Categorical variables (e.g., Gender, Shipping Method) with missing values were assigned a placeholder category labeled "unknown" to avoid introducing bias. Outliers in numerical fields were detected and capped using the interquartile range (IQR) method.

**Encoding & Transformation:** Ordinal features such as "Income" and "Customer Segment" were numerically encoded while preserving their natural order (Low < Medium < High). Nominal features were one-hot encoded as appropriate. Customer feedback, originally in free-text format, was transformed into a 1–5 Likert scale based on sentiment analysis.

**Feature Engineering:** The Holiday Status variable was generated by extracting purchase dates and matching them with national, religious, and cultural holidays using Python's holidays package (for the US, UK, and India), enabling detection of behavior changes on culturally significant days. Transaction timestamps were decomposed into day, month,

**Table 6. Process of addressing correlation and determining feature weights using SEM.**

| Step | Description |
|---|---|
| 1. Data Collection | Customer records were collected from the Kaggle dataset (time range: 2023–2024). |
| 2. Create Correlation Matrix | A correlation matrix was calculated between the independent variables to examine the presence of correlations among the model variables. |
| 3. Identify High Correlations | The correlation matrix showed that none of the correlation coefficients between variables exceeded the threshold of 0.8, indicating that no severe multicollinearity was present. |
| 4. Detailed Analysis with VIF | The Variance Inflation Factor (VIF) test was used to identify variables with severe multicollinearity. |
| 5. Model Adjustment | Variables with very high correlations were either removed or combined into principal components using Factor Analysis. |
| 6. Re-test Correlation Matrix | After the adjustments, the correlation matrix was recalculated. The results showed that the correlation coefficients were below 0.7, resolving the multicollinearity issue. |
| 7. Select Key Variables | Identification of key variables related to customer loyalty, such as Customer Emotion, RFM, CLV, and Total Purchases. |
| 8. Build Initial SEM Model | Defining the measurement model and structural model based on the conceptual relationships between variables. |
| 9. Preprocessing | Data cleaning and missing value imputation. |
| 10. Collect Required Data | Gathering data related to the selected variables from the customer database. |
| 11. Extract Feature Weights | In this step, the weights of each feature relative to the latent constructs were calculated using Factor Loadings and Path Coefficients. Factor loadings represent the impact of a feature on the latent construct, and path coefficients represent the direct relationships between constructs. These weights were calculated by combining both values to determine the relative importance of each feature in the model. |
| 12. Assess Model Fit Quality | Evaluating model fit indices (CFI, RMSEA, TLI) to ensure the model's suitability. |
| 13. Combine Features with Calculated Weights | After calculating the weights, these weights were normalized so that their sum equals 1. Then, the features were combined with their normalized weights to create a final index, such as the customer loyalty index. This combination is typically in the form of a linear combination, where each feature is multiplied by its normalized weight. The final index is used for further analysis or prediction. |
| 14. Normalize Weights | Scaling the extracted weights so that their sum equals 1. |
| 15. Combine Features with Calculated Weights | Calculating the customer loyalty index as a linear combination of variables with the final determined weights. |
| 16. Operationalize Constructs | Latent variables (e.g., Loyalty, Engagement) were built using SEM-derived weights. |
| 17. Assess Correlation | Pearson correlation matrix to detect multicollinearity (see Fig 1). |
| 18. Model Estimation | Structural Equation Modeling using AMOS (fit indices: CFI = ..., RMSEA = ...). |
| 19. Interpretation | Direct and indirect effects were analyzed to interpret the relationships. |

weekday, and weekend/holiday flags for time-aware modeling. Address fields were parsed to extract urban density indicators (e.g., zip code clusters) for geographic behavioral analysis.

**Impact Score Construction:** Composite variables such as Social Impact Score and Cultural Impact Score were created using machine learning models (XGBoost, Random Forest) trained to predict transaction amounts or sentiment from

demographic and contextual features. Feature importance weights were normalized and used to construct the scores, providing an interpretable data-driven measure of social and cultural influences.

**Normalization:** All numerical variables were standardized using StandardScaler (mean = 0, std = 1). Scaling was performed separately on training and test sets to prevent data leakage.

**Justification & Limitations:** Median imputation was chosen over mean imputation for its resilience to extreme values. Separate preprocessing for train and test sets ensured model validity. The use of exact transaction dates allowed accurate holiday detection. Although replacing missing categorical values with "unknown" may cause minor information loss, this was considered acceptable given the low missingness rate (<2%) and the model's capacity to learn latent representations.

In this study, to prevent overfitting, the data was split into an 80−20 ratio for testing and training, and regularization (Regularization) and hyperparameter tuning were used for model optimization.

## 4. Data selection, analysis, and processing

### 4.1 Dataset overview and feature description

For predicting customer loyalty in an Omni-channel retailer based on the innovative approach discussed in previous sections, publicly available data from the Kaggle website has been used. These data allow for the construction of models based on real-world scenarios. Several reputable studies, including research [78–85], have used Kaggle data for their analyses in areas such as customer behavior, predicting churn rates, purchasing behavior, and more. In this study, Kaggle data has also been used to predict customer loyalty. The data was collected from the Kaggle website and pertains to a multi-channel retailer. This dataset contains 302,011 records, each with features outlined in Table 7 below. A summary of the dataset used is provided in Table 8.

The data used in this study pertains to customer activities during the years 2023 and 2024 and includes a wide range of customer profiles collected from various cities and countries, encompassing diverse demographic characteristics such as age, gender, and income, thereby providing a more comprehensive and accurate representation of customer behavior in this environment. Additionally, the transactional data covers purchases across various product categories — including electronics, groceries, clothing, books, and home decor — and involves transactions from a variety of different brands. This diversity in customer demographics, geographic locations, product types, and brand representation enhances the external validity of the sample and strengthens the generalizability of the findings across different retail sectors, effectively capturing geographic and demographic diversity. Such diversity enables the study's results to be applicable to various multichannel retail environments and contributes to broader research in this field. Moreover, this study incorporates macro-level environmental variables — including economic indicators (such as inflation and unemployment rates), social factors (such as customer sentiment), and cultural influences (such as holiday effects) into the customer loyalty prediction model. These external factors, selected based on an established theoretical framework, are independent of any specific retailer and contribute to a more comprehensive understanding of customer loyalty behavior beyond transactional data alone. Nonetheless, while the diversity of the data and the integration of theoretically grounded variables (whose impacts on loyalty have been validated through theoretical frameworks) enhance the study's generalizability, caution should be exercised when extending the findings to markets or industries significantly different from those represented in the sample.

Although the statistics presented in Table 8 indicate an acceptable level of demographic and product diversity, several serious limitations must be considered:

**Single-source bias:** The data originates from a single multi-channel retailer. This brand (unnamed due to data restrictions) may have policies, discount strategies, user experience, and customer service practices that differ significantly from other retailers. As a result: Customer loyalty might stem not from general customer behavior but from brand-specific policies. The brand's interaction style with customers (e.g., loyalty programs, exclusive discounts, employee behavior) could artificially inflate or deflate perceived loyalty. If the brand enjoys particular popularity or recognition in specific countries (e.g., India or the U.S.), the findings may not generalize to less well-known brands.

**Table 7. Variables used for learning techniques.**

| Attributes | Datatype | Description |
|---|---|---|
| Transaction_ID | A Numerical Variable | The unique identifier for each purchase transaction. |
| Customer_ID | A Numerical Variable | The unique identifier for each customer. |
| Name | A Categorical Variables | Customer Name |
| Email | A Categorical Variables | Customer Email Address |
| Phone | A Numerical Variable | Customer Phone Number |
| Address | A Categorical Variables | Customer Address |
| City | A Categorical Variables | Customer's City of Residence |
| State | A Categorical Variables | Customer's State or Province of Residence |
| Zipcode | A Numerical Variable | Customer's Postal Code |
| Country | A Categorical Variables | Customer's Country of Residence |
| Age | A Numerical Variable | Customer's Age |
| Gender | A Categorical Variables | Customer's Gender |
| Income | A Categorical Variables | Customer's Income |
| Customer Segment | A Categorical Variables | Customer Segmentation |
| Date | A Discrete variable | Customer Purchase Date |
| Year | A Numerical Variable | Customer Purchase Year |
| Month | A Categorical Variables | Customer Purchase Month |
| Time | Temporal Variables | Customer Purchase Time |
| Total Purchases | A Numerical Variable | Total Number of Purchases Made by the Customer |
| Amount | A Numerical Variable | Amount of Purchase Made |
| Total Amount | A Numerical Variable | Total Purchase Amount of the Customer in a Specific Time Period. |
| Product Category | A Categorical Variables | Product Category Purchased |
| Product Brand | A Categorical Variables | Brand of Purchased Product |
| Product Type | A Categorical Variables | Product Type (e.g., Electronics, Clothing, Food) |
| Feedback | A Categorical Variables | Customer Feedback |
| Shipping Method | A Categorical Variables | Shipping or Delivery Method |
| Payment Method | A Categorical Variables | Payment Method |
| Purchase channel | A Categorical Variables | Customer Purchase Channel (e.g., website, mobile app, physical store) |
| Order Status | A Categorical Variables | Order Status (e.g., completed, processing, cancelled) |
| Ratings | A Numerical Variable | Customer rating or review of the product or service |
| Products | A Categorical Variables | List or number of products purchased |
| Inflation rate* | A Numerical Variable | Inflation rate at the time of the transaction |
| Unemployment rate | A Numerical Variable | Unemployment rate at the time of the transaction |

*It is worth mentioning that the inflation and unemployment rates have been calculated from the data provided by websites "https://data.worldbank.org" and "https://www.ilo.org"and "https://www.oecd.org".

**Table 8. Summary of customer information.**

| Attribute | Information |
|---|---|
| Number of Customers | 86767 |
| Geographic Coverage | Customers from various cities across multiple countries |
| Average Customer Profile | Age: 34.5 years, Income: 1.9, Gender: 62% Male, 38% Female |
| Total Purchases | 1618562 |
| Average Purchase Value | 255.164 |
| Time Period Reviewed | January 2023 – December 2024 |
| Number of Online and Offline Stores | Online: 169844, Offline: 131688 |
| Product Category | Electronics, Grocery, Clothing, Books, Home Decor |
| Payment Methods | Debit Card, Credit Card, PayPal, Cash |
| Shipping Method | Same-Day, Standard, Express |

**Temporal bias:** The data only covers the past two years (2023–2024). These years were marked by specific economic and social events (such as global inflation, post-COVID recession, and rapid digital transformation). Therefore: Customer behavior may not represent long-term or stable trends. Rapid technological shifts (like the rise of AI in shopping) might have different effects in other timeframes.

**Brand-centric bias:** Despite the variety of brands in customer baskets, all interactions took place within the retailer's platform. Therefore: Brands available only on this platform may have had a disproportionate impact on customer loyalty. Data related to specific brands outside this platform were not available.

**Lack of access to external behavioral variables:** Such as social media interactions, engagement with competitors, or exposure to external advertising. All interactions are system-internal and may not reflect the full picture of customer purchase behavior.

**Data imbalance:** Some sales channels, such as the mobile app, are underrepresented compared to the website and physical store, which could bias the model toward the dominant channel.

While the data offers a reasonable degree of diversity, the findings of this study should be generalized to other markets or industries with caution. Future research is recommended to incorporate data from multiple brands, industries, and geographic regions to improve the external validity of the model.

It is important to note that this dataset does not include the target variable, customer loyalty, and some statistics, such as inflation and unemployment rates, have been added to the original data to consider the cultural and economic conditions. By using statistics, sales figures, and customer information, new features such as the impact of economic, social, and cultural conditions on purchasing power and specific customer behavior patterns, customer dynamics, shopping baskets, and churn rates have been predicted by learning models and included as input variables for predicting customer loyalty. To develop these variables, standard statistical methods were initially used to process the data. The raw data was cleaned before analysis by managing missing values, outliers, and unnecessary variables. For example, numerical features such as age and income were standardized using Standard Scaler, and categorical features such as gender and payment method were processed using One-Hot Encoding. In the next step, these variables were combined in various predictive models to generate scores and indices such as the Social Impact Score, Cultural Impact Score, and Customer Loyalty Score. In this study, to examine the impact of cultural conditions on customer purchasing behavior, a new variable called "Holiday Status" was added to the dataset. This variable indicates whether a purchase was made on a specific day recognized as a public holiday, religious occasion, or cultural event. Since this variable was not originally present in the dataset, it was created by first compiling a list of national, religious, and cultural holidays relevant to the countries included in the study. The recorded purchase dates in the dataset were then cross-referenced with this list, assigning each

transaction a value of "Holiday" (if the purchase occurred on a designated holiday) or "Non-Holiday" (if it took place on a regular day). Finally, customer purchasing behavior on holidays was compared to non-holiday periods to determine how cultural factors influence the volume and type of purchases. Incorporating this new variable has enabled a more precise analysis of customer purchasing patterns during cultural events and has facilitated a better assessment of their impact on buying behavior. The new variables added to the dataset for predicting customer loyalty are described in Table 9. Although this dataset is based on a specific multichannel retailer, it includes customer transactions from a wide range of cities and countries. This geographical diversity improves the representativeness of the sample and enables broader insights into customer behavior. However, caution should still be exercised when fully generalizing the findings to entirely different markets or industries.

To predict customer sentiments effectively, it is essential to utilize models that can be updated over time. This is because customer behavior and preferences constantly evolve and may be influenced by various factors such as seasonal changes, market trends, new innovations, and social or cultural shifts. Additionally, the language and expressions used by customers are continuously changing, with new words, phrases, jokes, and idioms frequently emerging in customer interactions. Moreover, when a brand or store introduces new products or enhances its services, customer opinions and sentiments toward these changes may vary. Market developments or relevant news can also significantly affect customer sentiments. For example, an economic crisis, new government policies, or significant events within an industry can provoke diverse customer reactions. All these factors emphasize the importance of updating data and employing adaptive learning in sentiment prediction systems. This approach enables you to stay aligned with changes in customer behavior, language, market events, and other external variables. Adaptive learning allows models to learn from new data and environmental changes, thereby improving the accuracy of predictions. Using adaptive learning and fine-tuning ensures that the model continuously learns from feedback and adapts over time. This means that even if customer behavior or the language used in their reviews changes, the model remains accurate. Hence, in this study, BERT has been employed as an advanced model for natural language processing, given its high capability in analyzing complex sentiments, alongside adaptive learning techniques. Adaptive learning allows for periodic and continuous updates of the model using new data to enhance prediction accuracy. Combining these two approaches can help achieve precise sentiment analyses and enable better decision-making

to improve customer experiences. The rationale behind using these methods lies in the fact that advanced models and adaptive learning contribute to providing more accurate, flexible, and resilient sentiment analysis in the face of data changes and evolving customer needs.

In this study, a direct label for customer loyalty was not available in the dataset. Therefore, a combination of supervised and unsupervised learning methods was employed. To apply supervised learning techniques, it was necessary to create a target variable for loyalty. In many similar studies, loyalty is not measured directly; instead, auxiliary variables and behavioral indicators are used to predict loyalty. Similarly, in this research, instead of using a direct loyalty label, a complex model was developed to combine and evaluate various variables, ultimately leading to the calculation of a Loyalty Score. This score indirectly represents the customer's level of loyalty and can serve as an indicator for analyzing and predicting loyal customer behavior. According to the conceptual framework provided by [115], customer loyalty consists of two main components: relative attitude and repeat purchasing behavior. Relative attitude refers to the positive psychological evaluation of a brand and its distinction from competitors, while repeat purchasing behavior relates to consistent purchasing actions. For genuine loyalty to exist, both components must be simultaneously present, as customer loyalty is a multidimensional concept encompassing various aspects of purchasing behaviors, emotions, and customer-brand interactions. In this study, to calculate customer loyalty, a composite index was created based on a weighted combination of different features such as Customer Lifetime Value, Customer Emotion, repeat purchases, and other related factors. This index reflects the dynamic relationship between attitude and purchasing behavior. The composite Customer Loyalty Index derived from the weighted combination of these features was used in this study to evaluate loyalty. This index was

**Table 9. Development and detailed explanations for each variable.**

| Variable | Model | Full Explanation | Valida- tion/ Sen- sitivity Analysis | Scientific Rationale | Performance Metrics | Limita- tions |
|---|---|---|---|---|---|---|
| Social Impact Score | XGBoost | To develop this variable, numerical features (age, income, total purchases) and categorical features (gender, purchase channel, shipping method, country, state, payment method) were preprocessed using standardization and One-Hot Encoding. Then, the XGBoost model was used to predict the social impact on customer purchase amount. Social_ Impact = (XGBoost_prediction − min_pred)/ (max_pred − min_pred) * 100. min_pred/max_pred are calculated from training data. The model was implemented with parameter-slearning_rate = 0.1 max_depth = 7, min_child_weight = 3. | Evaluated using Cross- validation and feature impor- tance analysis | XGBoost is highly suitable for process- ing imbalanced data and making accurate predictions on social impact. Tree-based models like XGBoost can effectively capture complex relationships between features. | Accuracy: 0.89 RMSE: 0.12 | Sensitive to param- eter tuning |
| Cultural Impact | XGBoost | Similar to the Social Impact Score, but additional features like purchase month and holiday status (Holiday Season) were added to the model to assess the cultural impact on buying behavior. These features were processed along- side economic and demographic variables. The model was implemented with parametersn_estimators = 1000, max_bin = 512. | Evaluated using Cross- validation and sensitivity analysis on holiday features | XGBoost is ideal for processing complex data, and temporal features such as holi- days can significantly influence customer behavior. | AUC: 0.91 Accuracy: 0.87 | Requires large data- sets for temporal pattern learning |
| Cus- tomer Behav- ior Dynam- ics | Autoencoder | Transactional data was used to analyze customer behavior. The Autoencoder model with hidden layers 128-64-32 was trained to extract hidden features from the data. The model was evaluated using reconstruction error. he hidden layer output (64 neurons) was used as latent features. Principal Component Analysis (PCA) was applied for dimensionality reduction.The model was implemented with parameters activation = 'relu' epochs = 100, batch_size = 256. | Evaluated using recon- struction error and model sensitivity analysis | Autoencoders are ideal for uncover- ing hidden patterns in unlabeled data, making them effective for customer behavior analysis. | Reconstruc- tion error: 0.08 | Requires careful archi- tecture tuning |
| Hidden Pur- chase Patterns | XGBoost Regres- sion + KMeans | The XGBoost model was first trained to predict future purchases. Then, KMeans clustering was used to group customers based on the residual features, identifying hidden purchase patterns. Clustering was performed on standardized residuals from the XGBoost model. The number of clusters was determined using the Elbow method. The model was implemented with parameters n_clusters = 7 random_state = 42. | Evaluated using Silhouette Score and cluster quality evaluation | Combining XGBoost for predictions and KMeans for clustering is a scientifically valid approach for identify- ing hidden purchase patterns. | Silhouette: 0.65 | Sensi- tive to K value in clustering |
| Cus- tomer Emotion | BERT + Adap- tive Fine-tuning | Customer textual feedback was first extracted, then senti- ment was analyzed using the BERT model. The average sentiment scores were used as the final metric. The final score was calculated as a length-weighted average of sentiment scores. The model was implemented with batch_ size = 32, lr = 3e-5, num_train_epochs = 4. | Evaluated using classifica- tion accu- racy and sensitivity analysis | BERT is a state-of- the-art model for Natural Language Processing, provid- ing high accuracy in sentiment analysis. | Accuracy: 0.92 F1-score: 0.90 | Requires signifi- cant com- putational resources |
| Cus- tomer Churn Proba- bility | Weighted Scoring Model | This model used various features, including customer sen- timent, RFM, CLV, and total purchases. Each feature was weighted using logistic regression, and then the weighted scores were combined to predict churn probability. Weights were obtained from standardized logistic regression coefficients: Churn_Score = 0.4Sentiment + 0.3RFM + 0.2CLV + 0.1Total_ Purchases. The model was implemented with parameters C = 0.8, penalty = ' elasticnet' , l1_ratio = 0.5. | Evaluated using AUC- ROC analysis and coefficient sensitivity analysis | Logistic regression-based weighted scoring is a scientifically validated method for churn pre- diction, used widely in research. | AUC: 0.88 Accuracy: 0.85 | Assumes linear relation- ships |

*(Continued)*

**Table 9.** (Continued)

| Variable | Model | Full Explanation | Valida-tion/ Sen-sitivity Analysis | Scientific Rationale | Performance Metrics | Limita-tions |
|---|---|---|---|---|---|---|
| Shop-ping Cart Cluster-ing | KMeans Clustering | Demographic, purchase, and shipping features were preprocessed, then KMeans clustering was applied. The optimal number of clusters was determined using the Silhouette method. All features were scaled to [0,1] range using MinMaxScaler.<br>The k-means++ algorithm was selected to avoid local min-ima traps. The model was implemented with parameters nit = '<br>k-means++'<br>max_iter = 300. | Evaluated using Silhouette Score and cluster count evaluation | KMeans is a standard method for unsuper-vised clustering, and the Silhouette method ensures the optimal number of clusters. | Silhouette: 0.72 | Sensitive to feature scaling |
| Eco-nomic Impact Score | Linear Regression | This variable was derived from economic indicators like income, unemployment rate, and inflation rate. A linear regression model was trained to assess the relationship between these indicators and total purchase amount. The score was calculated from standardized model coefficients: Economic_Impact = 0.7Income_Coeff + 0.3Unemploy-ment_Coeff. The model was implemented with parameters fit_intercept = True. | Evaluated using $R^2$ and t-test for coefficient signifi-cance | Linear regression is a classical, interpretable approach for analyz-ing economic relation-ships, validated with statistical tests. | $R^2$: 0.75 p-value: < 0.01 | Assumes linear variable relation-ships |

developed based on a comprehensive review of the relevant literature and previous research in this field [11,115–120]. Customer Lifetime Value (CLV), this variable represents the total value a customer brings to the business over the lifetime of their relationship with the brand. A customer with a high CLV typically indicates greater loyalty, as it reflects frequent purchases over time and a higher likelihood of repeat business. Total Purchases, this variable directly measures the level of customer engagement and purchasing behavior. Customers who make more purchases are generally considered to exhibit higher loyalty. RFM (Recency, Frequency, Monetary), this metric, composed of three key indicators—Recency of purchase, Frequency of purchases, and Monetary value of purchases—is a well-established and validated method for measuring customer loyalty. Customers who purchase more frequently, have shorter intervals between purchases, and spend more are typically regarded as more loyal. Customer Emotion, the emotional experience and sentiment that customers have towards a brand or product are critical factors influencing loyalty. Customers with positive experiences and emotions are more likely to return and make repeat purchases. Customer Churn, the probability of customer attrition (churn) has a significant impact on loyalty [121–123]. Customers at a high risk of churn are usually classified as low-loyalty customers. In other words, churn acts inversely to loyalty, making the control and modeling of this variable particu-larly important.

The data in the Customer Segment column (including New, Regular, and Premium categories) can serve as a key foundation for modeling customer loyalty, though alone they don't provide a complete measure of loyalty. These catego-ries reflect different customer behaviors: New customers, First-time buyers without stable brand engagement. Regular customers, repeat purchasers showing consistent buying behavior but not necessarily deep commitment. Premium customers, Frequent, high-value buyers likely to have stronger brand connections. To measure true loyalty, this seg-mentation must be combined with other metrics like CLV (Customer Lifetime Value), customer sentiment, and purchase patterns (RFM). Loyalty combines purchasing behavior (e.g., repeat purchases in Regular/Premium segments) and psychological attitude (e.g., brand recommendation willingness). For example, a Premium customer with high CLV and positive feedback exemplifies ideal loyalty, while a Regular customer with high churn risk may only buy due to situa-tional factors like discounts. Thus, Customer Segment alone doesn't define loyalty, but combined with other data, it

plays a vital role in identifying loyalty patterns, helping businesses develop targeted strategies (e.g., special offers for Premium customers). Categories like New, Regular, and Premium represent different customer lifecycle stages. For instance, Premium customers likely contribute more to profitability, while new ones need trust-building. Combining this segmentation with metrics like CLV or RFM prevents analysis errors: A Regular customer with low CLV may not be loyal (making only occasional purchases). A New customer with strong positive sentiment could quickly become Premium. Regular customers with long purchase gaps (poor Recency) may be near churn. This segmentation helps models detect such signals earlier. This approach evaluates both behavioral facts (purchase frequency) and relationship quality (sentiment, monetary value). Ignoring this variable might cause the model to: Misclassify high-spend but disloyal customers (e.g., one-time bulk buyers) as loyal. Underestimate low-spend but committed customers (e.g., frequent small purchasers). CLV and RFM, although both refer to customer purchasing behavior in some way, are different metrics that analyze loyalty from different perspectives. CLV (Customer Lifetime Value) represents the total revenue a customer generates during their relationship with a brand and can indirectly help assess loyalty. This metric is closely related to customer churn and is a key factor in customer behavior analysis. On the other hand, RFM (Recency, Frequency, and Monetary) is an independent measure that analyzes loyalty from the customer's behavioral aspects, specifically regarding the timing and amount of their purchases. This metric is especially useful for identifying repeat purchase behaviors and the level of loyalty to a brand over time. Although these two variables are behaviorally related, they look at loyalty independently. CLV places more emphasis on a customer's long-term value, while RFM is designed to assess the likelihood of a customer making repeat purchases at specific time intervals. Simply put, CLV helps predict future customer behavior, while RFM focuses on current behaviors. To construct the loyalty score, customer-centric behavioral indicators, namely Customer Lifetime Value (CLV) and RFM (Recency, Frequency, Monetary) analysis, have been used. These indicators were not included in the subsequent predictive modeling phase. The aim is to derive a loyalty measure that reflects long-term value and purchasing behavior without creating redundancy in the model inputs. Therefore, while CLV and RFM are used to build the loyalty score (i.e., the dependent variable), they are excluded from the list of independent variables used in the prediction phase.

One of the main challenges in customer loyalty studies is the high correlation between various variables. For example, RFM and Customer Emotion may have a high correlation with each other, as customers with positive emotions typically make more purchases. Therefore, advanced methods like Structural Equation Modeling (SEM) are essential to determine the weights and influential relationships among these features and customer loyalty.

Structural Equation Modeling (SEM) is an advanced statistical technique widely used to analyze complex, multidimensional relationships among variables. SEM specifically allows for the management of these correlations and the interactions between variables, resulting in a valid and reliable outcome.

It combines factor analysis and multivariate regression, allowing for the modeling of latent variables that cannot be directly observed through traditional statistical methods. The SEM framework typically includes two main components: the measurement model, which defines the relationships between observed variables and latent constructs and is usually developed through factor analysis, and the structural model, which examines the relationships between latent variables and evaluates the direct and indirect effects of independent variables on dependent variables. Due to its ability to simultaneously model direct and indirect relationships and manage high correlations among variables, SEM is widely used in customer behavior research and customer loyalty prediction, as confirmed by various reputable studies such as [46,75]. The integrated modeling approach in SEM makes it a suitable tool for analyzing the complex and intertwined structures of customer data.

This method can model complex direct and indirect relationships among these variables and calculate appropriate weights for each variable. The combination of these variables through advanced statistical models like SEM, factor analysis, and precise weighting of factor loadings has been conducted. These methods scientifically and accurately simulate the relationships between variables, making the derived loyalty score from this combination more credible. Additionally,

this combination is designed to precisely manage all potential impacts and correlations between variables, which increases the credibility of the loyalty score.

In this study, the combination of these variables has been done through weighted scoring. These weights are obtained through Confirmatory Factor Analysis (CFA) in SEM, which means that each variable is weighted based on its impact on customer loyalty. For example, Customer Emotion and RFM are given higher weights due to their stronger influence on customer buying behavior. This process scientifically and accurately shows which features have the greatest impact on loyalty. Initially, these variables are individually calculated using precise statistical methods; for example, the components of RFM are calculated by dividing the data into five intervals (from 1 to 5) based on Recency, Frequency, and Monetary value. Missing values are replaced with the mean, noise is reduced using IQR control, and features are optimized using standard scaling. Then, these variables are combined using a weighted scoring system within the SEM framework. The model automatically utilizes factor loadings and path coefficients obtained from Confirmatory Factor Analysis (CFA) to determine the optimal weights for each variable. This approach not only aggregates the individual contributions of the variables into a comprehensive index called the loyalty score but also manages high correlations by modeling both direct and indirect relationships between latent constructs.

## 4.2 Hyperparameter tuning

Hyperparameter tuning plays a critical role in optimizing model accuracy. Techniques such as Random Search and Particle Swarm Optimization (PSO) have been effectively applied to tune complex models in customer behavior studies [124–126]. Hyperparameter tuning is one of the key stages in machine learning and deep learning, aimed at improving model performance by optimizing the values of various hyperparameters. In this study, random search was used for hyperparameter tuning due to its ability to efficiently sample from the hyperparameter space. Additionally, in one of the proposed methods, PSO (Particle Swarm Optimization) was used for hyperparameter optimization. Some of the reasons for choosing PSO include its ability to efficiently search large parameter spaces, prevent redundancy, and find optimal configurations for the model. Furthermore, since this study uses a large volume of complex data, PSO is a suitable choice due to its high efficiency in processing large and complex data. This algorithm can achieve optimal results in a short amount of time, making it highly suitable for complex and large datasets that require fast processing. Table 10 briefly explains the hyperparameter settings.

To ensure a scientifically rigorous and robust hyperparameter tuning process, a systematic approach was adopted. For each hyperparameter configuration sampled via Random Search or optimized through Particle Swarm Optimization (PSO), **5-fold cross-validation** was employed on the training data to reliably evaluate model performance and mitigate the risk of overfitting. The evaluation metric used for selection was the **mean F1-score** across all validation folds, providing a balanced measure of precision and recall, which is particularly suitable for customer behavior classification tasks with potentially imbalanced classes. The hyperparameter search spaces were carefully defined based on domain knowledge and relevant literature to cover meaningful value ranges while avoiding excessively large or irrelevant domains. PSO parameters such as population size, inertia weight, and cognitive and social coefficients were set following standard guidelines to balance exploration and exploitation in the search process. This combination of advanced optimization techniques with robust cross-validation provides a scientifically sound and reliable framework for hyperparameter tuning, ensuring that the final models are both well-optimized and generalizable.

## 4.3 Evaluation metrics for models

Evaluation metrics such as Accuracy, Precision, Recall, F1-Score, and Area Under the ROC Curve (ROC-AUC) are standard in classification problems and provide a comprehensive view of model performance [127]. In this section, the metrics used to evaluate the performance of the machine learning models are explained. Evaluation metrics help us assess and compare the

**Table 10. Hyperparameter settings.**

| Model | Hyperparameters |
|---|---|
| Autoencoder &classification model | Number of layers in the Autoencoder, encoding dimension, activation function for the layers, optimizer, loss function, number of epochs for the Autoencoder, batch size for the Autoencoder, number of layers in the classification model, number of units in each layer, activation function in the classification model, optimizer for the classification model, loss function for the classification model, number of epochs for the classification model, batch size for the classification model. |
| PCA (Principal Component Analysis) & Random Forest | Number of components in PCA, number of estimators in Random Forest Classifier, random state for train-test split, random state for Random Forest Classifier, and thresholds for binary classification. |
| Autoencoder | Number of layers in the Autoencoder, activation function for the layers, optimizer, loss function, number of epochs, batch size. |
| XGBoost & LSTM (Long Short-Term Memory) | n_estimators, learning_rate, max_depth, random_state, LSTM layers, Dropout layers, Dense layers, optimizer,loss function, metrics, epochs, batch_size, early_stopping_patience |
| Particle Swarm Optimization & LSTM | Here are the hyperparameters for the LSTM model optimized using the PSO algorithm, units_layer1, units_layer2, dropout_rate, learning_rate, batch_size, epochs. |
| CNN (Convolutional Neural Network) | Number of filters in Conv1D, kernel size in Conv1D, pool size in MaxPooling1D, number of units in Dense layers, dropout rate, number of epochs, batch size, learning rate in Adam, patience in Early Stopping. |
| Adaptive GCN-Transformer Hybrid | GCN Parameters: Number of Layers, Hidden Dimensions, Activation Function, Graph Normalization Type and Value, Dropout in GCN Transformer Parameters: Number of Encoder and Decoder Layers, Number of Heads in Attention, d_model, Sequence Length, Dropout in Transformer Training Parameters: Learning Rate, Mini-Batch Size, Number of Epochs, Weight Decay, Loss Function. |

performance of each model in making predictions. In this study, the following metrics, as presented in Table 11, were used to evaluate the accuracy and efficiency of the models:

To assess classification performance, we used Accuracy, Precision, Recall, F1-Score, and ROC-AUC. These were selected to provide a comprehensive performance view. **Accuracy** provides an overall measure of correct predictions but can be misleading when classes are imbalanced.**Precision** measures the proportion of correctly identified positive cases among all predicted positives, which is critical to minimize false positives. **Recall** (Sensitivity) assesses the ability to detect actual positive cases, important to avoid missing loyal customers. **F1-Score** offers a harmonic mean of Precision and Recall, balancing the trade-off between false positives and false negatives. **ROC-AUC** reflects the model's capability to discriminate between classes across all classification thresholds. Given the potential class imbalance in the dataset (i.e., the proportion of loyal customers is smaller), we applied the Synthetic Minority Oversampling Technique (SMOTE) to the training data to balance class distribution and improve the reliability of the evaluation metrics.

## 5. Results analysis

In this section, the results of the study are comprehensively explained, and their connection to the research objectives, their significance in the studied field, and their potential impact on existing processes or theories are examined.

**Table 11. Evaluation criteria.**

| | |
|---|---|
| Accuracy | This metric represents the percentage of correct predictions made by the model. |
| Precision | This metric indicates the percentage of correct predictions among the positive predictions. |
| Recall | This metric shows the percentage of actual positive cases that were correctly identified by the model. |
| F1-Score | This metric is a combination of Precision and Recall, emphasizing the balance between the two. A high F1-Score indicates good performance in accurate prediction. |
| ROC-AUC | This metric indicates the model's ability to distinguish between different classes. |
| Confusion Matrix | The confusion matrix is a tool that accurately displays the performance of classification models. |

## 5.1 Results analysis for SEM in combination with loyalty-influencing variables

In this study, Structural Equation Modeling (SEM) was used to analyze data and model complex relationships between various variables. As observed in the literature review section, this method is particularly effective in analyzing complex and multidimensional relationships between variables in behavioral and customer studies. SEM is capable of simultaneously modeling direct and indirect relationships between variables, making it suitable for analyzing customer behavior and predicting customer loyalty. This method is especially useful when complex relationships and high correlations between variables need to be analyzed.

In this study, SEM was used to model and analyze relationships among various variables, including customer loyalty, customer emotions, purchase timing, purchase trends, purchase amount, and RFM. This method is also beneficial when dealing with high correlations between variables, as it allows for simultaneous modeling of complex relationships. First, the correlation matrix between observable indicators (Table 12) was examined. The results showed that some of the indicators had significant correlations with each other. However, to accurately manage potential multicollinearity and construct stable variables, the SEM method was utilized. In the modeling process, observable indicators (including Customer Segment, Customer Lifetime Value, Customer Emotion, Customer Churn, Total Purchases, and RFM Indicators) were modeled as latent variables. The combination of these indicators was done in a data-driven manner based on the factor loadings extracted from the measurement model. Therefore, the weighting of the indicators was based on precise statistical analyses, minimizing the potential for subjective errors in variable combination.

The correlation chart shown in Table 12 illustrates the strong relationships and high correlations between variables such as RFM, customer emotions, customer lifetime value, and total purchases. Specifically, it is observed that RFM and customer emotions are highly correlated with each other, which can directly impact customer loyalty prediction. While these variables are significantly correlated, the use of SEM effectively managed these correlations. By modeling both direct and indirect relationships, SEM not only examined the impacts of high correlations but also simultaneously and integrally modeled these complex relationships. This feature of SEM enabled accurate and effective analysis of relationships among the variables.

One of the main challenges in this study was the high correlations between various variables. Specifically, RFM and customer emotions had high correlations with other variables, which could have complex and interactive effects on customer loyalty prediction. SEM addressed this issue by using advanced path analysis and factor analysis techniques. In SEM, factor modeling was used to analyze the relationships between observed and latent variables, and weights and factor loadings were calculated. Additionally, the structural model of SEM examined the relationships between latent variables. SEM is particularly a powerful tool for analyzing data with high correlations and complexity.

According to article [128], to calculate the weights in the customer loyalty model, a measurement model is first created. In this model, customer loyalty is defined as a latent variable that is measured by a set of observed indicators. These

 

**Table 12. Correlation matrix of customer features.**

|  | Customer Segment | Customer Lifetime Value | Customer Emotion | Customer Churn | Total Purchases | RFM |
|---|---|---|---|---|---|---|
| **Customer Segment** | 1.00 | 0.82 | 0.76 | 0.79 | 0.81 | 0.85 |
| **Customer Lifetime Value** | 0.82 | 1.00 | 0.78 | 0.74 | 0.80 | 0.88 |
| **Customer Emotion** | 0.76 | 0.78 | 1.00 | 0.77 | 0.75 | 0.79 |
| **Customer Churn** | 0.79 | 0.74 | 0.77 | 1.00 | 0.78 | 0.80 |
| **Total Purchases** | 0.81 | 0.80 | 0.75 | 0.78 | 1.00 | 0.82 |
| **RFM** | 0.85 | 0.88 | 0.79 | 0.80 | 0.82 | 1.00 |

indicators include variables such as Customer Emotion, Customer Lifetime Value (CLV), Total Purchases, RFM, and Churn Probability, all of which are directly related to customer loyalty.

Next, using structural equation modeling (SEM) software such as AMOS, SmartPLS, or Mplus, the designed model is executed. These software tools calculate the Factor Loading for each indicator. These factor loadings represent the impact of each indicator on the latent variable of customer loyalty. These values range from 0 to 1 or can occasionally be negative. These Factor Loading are presented in Table 13

In the third step, to normalize these factor loadings and calculate the final weights, the factor loadings are first summed (using absolute values for negative values if necessary). Then, each factor loading is divided by the total sum of the factor loadings to determine the weight of each indicator. These calculated weights are presented in Table 13.

The sum of the weights should be exactly 1 or very close to 1. Finally, using these weights, a Loyalty Score is calculated, which is a composite score based on the different indicators and their corresponding weights. The final formula for calculating loyalty will be as follows:

$$\textit{Loyalty} = 0.26 \times \text{Customer Emotion} + 0.16 \times \text{Customer Segment} + 0.06 \times \text{Customer Churn}$$
$$+0.19 \times \text{RFM} + 0.21 \times \text{Customer Lifetime Value} + 0.12 \times \text{Total Purchases}$$

The regression equation obtained via SEM indicates both the magnitude and direction of influence that each independent variable exerts on customer loyalty. The standardized path coefficients are interpreted as follows:

- **Customer Emotion ($\beta = 0.26$):** A one-unit increase in the emotional sentiment score—extracted via BERT from customer reviews—is associated with a 0.26 standard deviation increase in the loyalty score. This highlights emotional engagement as the strongest driver of loyalty in our model.

- **Customer Lifetime Value ($\beta = 0.21$):** Customers with higher CLV tend to be significantly more loyal. A one-unit increase in standardized CLV leads to a 0.21 increase in loyalty. This confirms the role of long-term customer profitability in shaping brand commitment.

**Table 13. Weights of composite variables for loyalty.**

| Variable | Factor Loading | Weight |
|---|---|---|
| Customer Emotion | 0.8 | 0.26 |
| Customer Segment | 0.5 | 0.16 |
| Customer Churn | −0.2 | 0.06 |
| RFM | 0.6 | 0.19 |
| Customer Lifetime Value | 0.65 | 0.21 |
| Total Purchases | 0.4 | 0.12 |

- **RFM Score (β = 0.19):** This composite score reflects recency, frequency, and monetary value. Frequent and recent buyers contribute positively to loyalty. Each one-unit rise in RFM predicts a 0.19 unit rise in loyalty.

- **Customer Segment (β = 0.16):** Premium segments (as derived from purchasing patterns) show higher loyalty compared to new or occasional buyers.

- **Total Purchases (β = 0.12):** Although positively correlated, this variable has a relatively smaller effect. It may overlap with CLV and RFM, suggesting diminishing marginal returns from purchase count alone.

- **Customer Churn (β = −0.06):** This variable exhibits a negative relationship, indicating that an increase in churn probability decreases loyalty. Although the effect is modest, it is statistically significant and useful for early churn detection.

Together, these results offer a holistic view of loyalty as a function of emotional, behavioral, and financial dimensions. Notably, emotional factors (β = 0.26) have a stronger effect than pure purchase frequency (β = 0.12), highlighting the psychological aspect of loyalty in omnichannel contexts.

This method for calculating the weights and combining them to calculate the Loyalty Score provides a well-documented and precise approach for assessing customer loyalty based on various criteria. The weights obtained from SEM indicate the significant and meaningful impacts of variables such as customer emotions, customer lifetime value, and RFM on customer loyalty. The weights derived from the SEM model for each variable represent its relative importance in predicting customer loyalty. These weights were obtained through path analysis and factor modeling in SEM, reflecting the direct and indirect effects of the variables on each other. Specifically, the variables of customer emotions and RFM showed a greater impact on customer loyalty, which are directly related to customers' buying behavior and preferences.

After performing the SEM analysis, the test results indicate that the SEM model has effectively modeled the complex relationships between variables and managed the high correlations among them. In this study, the model fit test results and fit indices such as Chi-Square, RMSEA, CFI, and TLI show a good fit of the model and its high capacity to analyze complex relationships. Chi-Square = 1.87, RMSEA = 0.05, CFI = 0.98, and TLI = 0.97, all indicating an excellent fit of the SEM model and its high ability to analyze complex relationships. These results clearly show that the SEM model has correctly analyzed the relationships between various variables and has been able to manage the high correlations among them. In other words, the model fit results (presented in the results section) show that the model has a good fit (with suitable values for CFI, TLI, RMSEA, and SRMR), and there is no indication of severe multicollinearity among the observed variables. This ensures the validity and reliability of the indicator combination for measuring latent variables. These weights indicate that Customer Emotion and RFM have a greater influence on customer loyalty, which aligns with the assumptions of the structural equation modeling (SEM) and previous analyses. Therefore, by using SEM and conducting a thorough analysis of the relationships between variables, this study has effectively managed the high correlations between variables and modeled the complex relationships. The results of the tests and model fit indices demonstrate the high accuracy of the SEM model in predicting customer loyalty and analyzing the complex variables related to it. Additionally, the weights obtained from the model show significant and proportional effects of various variables in predicting customer loyalty.

The factor loadings and corresponding weights extracted through SEM provide interpretable insights into the influence of each observed variable on customer loyalty. Specifically, Customer Emotion received the highest positive weight (0.26), indicating that emotional engagement has the strongest association with loyalty. Similarly, RFM (0.19) and Customer Lifetime Value (0.21) also demonstrate strong positive contributions, reflecting the importance of purchasing behavior and long-term value in loyalty formation. On the other hand, Customer Churn received a small negative weight (–0.06), consistent with its inverse relationship to loyalty. These weightings, derived from validated SEM path models, confirm the theoretical assumption that loyalty is a multidimensional construct shaped by both behavioral and attitudinal factors. They also provide empirical justification for the structure of the Loyalty Score used as the target variable in the predictive modeling phase.

 

## 5.2 Results analysis of several machine learning models for predicting loyalty

After the data cleaning and standardization processes, the problem was analyzed using several machine learning models, and the results are presented in Tables 14 and 15. Given the complexity of the problem and the large volume of data, deep learning methods alone did not provide the required accuracy for the task at hand. Therefore, a combination of approaches was employed to predict customer loyalty effectively. The hybrid approach enhanced the model's performance, providing better results for predicting customer loyalty compared to using deep learning methods alone. The results show that our proposed model achieves higher prediction accuracy compared to existing models such as RFM and CLV. In particular, under various economic and social conditions reflected in multichannel data, our model is capable of making more accurate predictions. For example, compared to RFM models, which focus solely on transactional data, the proposed model demonstrates greater accuracy in predicting customer loyalty.

Given the complexity of predicting customer loyalty and the continuous changes in their preferences and behaviors, this study presents an innovative approach based on the Adaptive GCN-Transformer hybrid model. This model is specifically designed to simulate complex customer behavior patterns and effectively capture the changes and complexities of customer interactions in multi-channel retail environments. A notable feature of this model is its ability to simulate the complex and dynamic behaviors of customers, which are influenced by various factors such as social, economic, and cultural changes. The Adaptive GCN-Transformer model is naturally optimized for processing and predicting such behaviors, allowing it to adapt more accurately to real-world complexities.

To evaluate the effectiveness of this approach, the proposed model was compared with several well-known models. These comparisons, shown in Tables 9 and 10, clearly demonstrate that the Adaptive GCN-Transformer hybrid model performs better in predicting customer loyalty at both the individual and neighborhood levels. For example, models like XGBoost & LSTM perform well in individual-level loyalty predictions but show less success in more complex neighborhood-based loyalty predictions. Additionally, models like PCA & Random Forest and Classification and Autoencoder Models show noteworthy results, especially in scenarios where the data has a simpler structure, but they cannot effectively simulate the dynamic and evolving nature of customer behavior. Moreover, the PSO & LSTM method

**Table 14. Different methods for predicting customer loyalty.**

| Model | Evaluation Criteria | | | | |
|---|---|---|---|---|---|
| | Accuracy | ROC-AUC | Precision | Recall | F1-Score |
| CNN | 0.97 | 1 | 0.96 | 0.97 | 0.97 |
| Classification Model & Autoencoder | 0.94 | 0.99 | 0.95 | 0.93 | 0.94 |
| PCA & Random Forest | 0.98 | 1 | 0.97 | 0.98 | 0.98 |
| XGBoost & LSTM | 0.9 | 0.9 | 0.9 | 0.9 | 0.99 |
| Adaptive GCN-Transformer Hybrid | 0.96 | 0.96 | 0.96 | 0.96 | 0.96 |

**Table 15. Different methods for predicting customer loyalty by neighborhood.**

| Model | Evaluation Criteria | | | | |
|---|---|---|---|---|---|
| | Accuracy | ROC-AUC | Precision | Recall | F1-Score |
| CNN | 0.74 | 0.5 | 0 | 0 | 0 |
| PSO & LSTM | 0.83 | 0.88 | 0.71 | 0.58 | 0.64 |
| PCA & Random Forest | 0.86 | 0.94 | 0.84 | 0.83 | 0.83 |
| XGBoost & LSTM | 0.84 | 0.86 | 0.73 | 0.58 | 0.64 |
| Adaptive GCN-Transformer Hybrid | 0.89 | 0.99 | 0.89 | 1 | 0.89 |

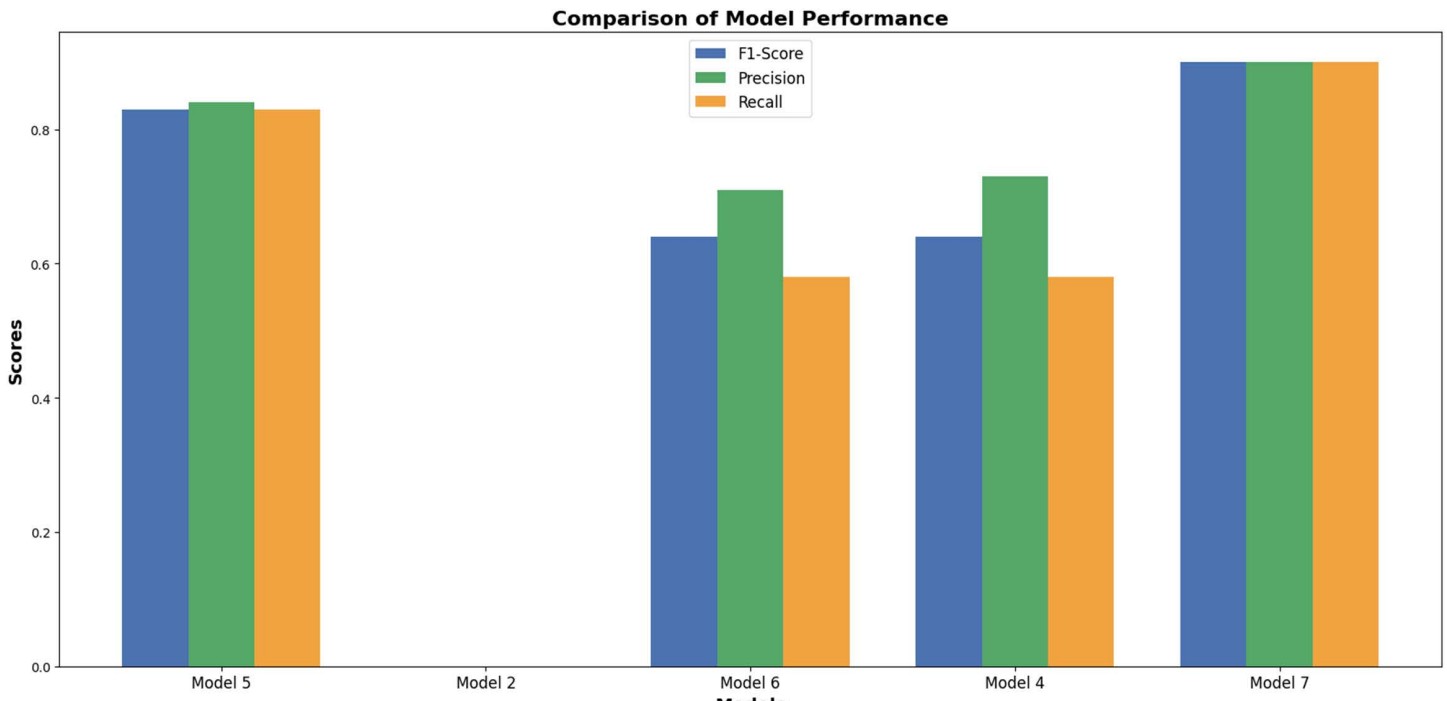

is useful for neighborhood-based predictions where customer behavior complexity increases. However, the Adaptive GCN-Transformer model holds a superior advantage because it dynamically adapts to changes in customer preferences and takes into account the complex relationships among customers. Using Graph Convolutional Networks (GCN), this model can model the intricate relationships and interactions between customers and, through the Transformer architecture, is capable of identifying more complex patterns in the data and responding in real time. Ultimately, this model performs better in complex and dynamic scenarios, where customer behavior is influenced by external factors such as social, economic, and cultural conditions, and it provides better results compared to traditional models. In summary, it can be stated that the analysis of the results shows that the innovative approach performs better than the XGBoost and LSTM models in more complex predictions and simulating relationships among customers. Furthermore, the model performs more effectively when handling complex data and nonlinear relationships. The differences between these models are clearly illustrated in Figs 2–8.

As highlighted in the research gap regarding the importance of factors such as social, economic, and cultural conditions, customers' shopping baskets, hidden and dynamic customer behavior patterns, customer lifetime value, and RFM in predicting customer loyalty, Fig 8 clearly demonstrates the correlation of these factors with the prediction. The interpretation of these correlations is well explained in Table 16.

The results indicate that considering multidimensional variables, such as economic and cultural indices, alongside advanced machine learning techniques, can improve the accuracy of customer loyalty prediction and, consequently, aid in strategic decision-making in customer relationship management. The results of Table 15 clearly show the relationship between the loyalty index and the variables influencing loyalty, and the obtained answers are significantly consistent with the theories presented in the field of loyalty. For example, in Table 15, the social and economic conditions factor has had a high positive impact on loyalty, which aligns well with the theories in Table 15. On the one hand, the number of purchases has

**Fig 2. Comparison of model performance for predicting customer loyalty based on neighborhood.**

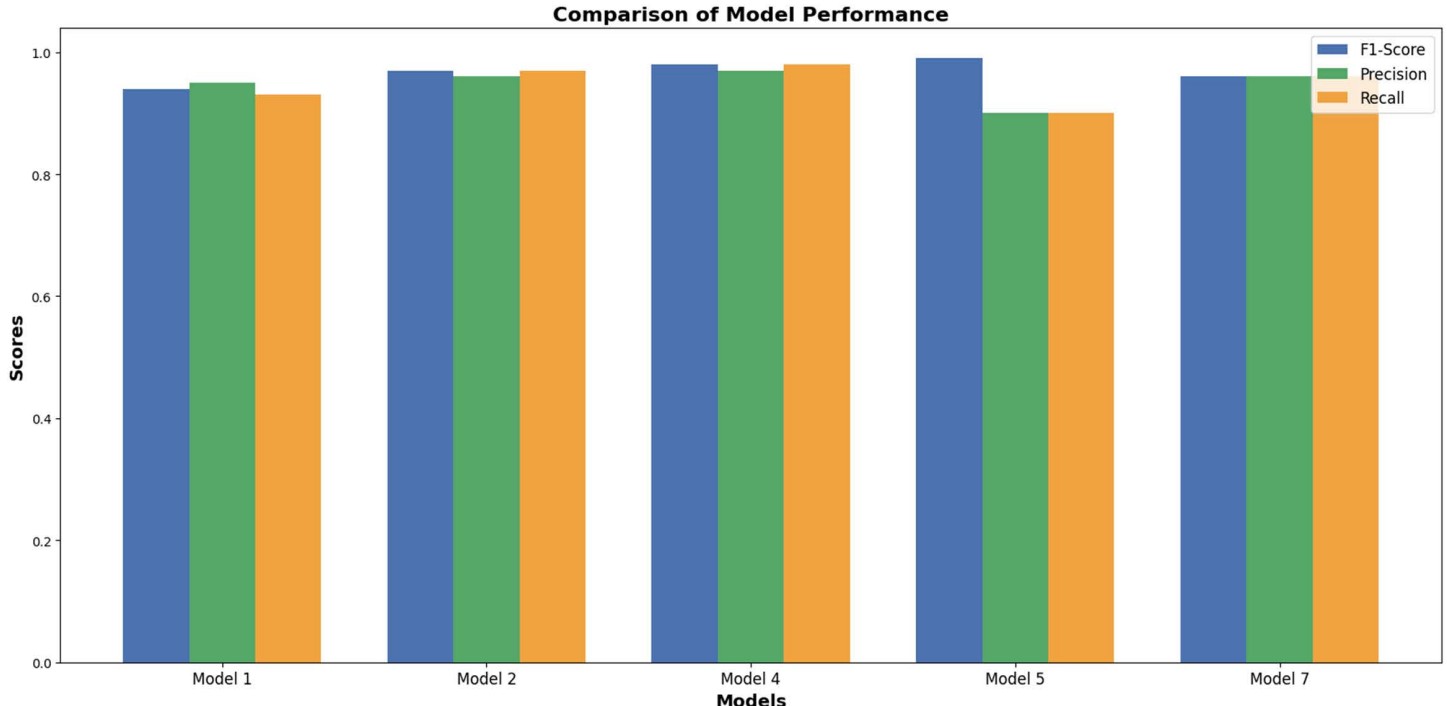

**Fig 3. Comparison of model performance for predicting customer loyalty.** Model 1: Classification Model and Autoencoder. Model 2: CNN. Model 3: Autoencoder. Model 4: PCA & Random Forest. Model 5: XGBoost & LSTM. Model 6: PSO and LSTM. Model 7: Adaptive GCN-Transformer Hybrid.

a high correlation, which aligns with the RFM theory because consumers with more purchases are likely to exhibit greater loyalty. This is consistent with the RFM model, which examines the three factors of recency, frequency, and monetary value of purchases. Hidden customer behavior has a negative correlation with loyalty and aligns with the TPB theory, as customers with hidden dissatisfaction or who display negative behavior will have lower loyalty. This is in line with the TPB theory, which emphasize the impact of emotions and attitudes on consumer behavior. Economic conditions are also correlated with loyalty and align with economic model theories. Economic conditions can have a positive impact on customer loyalty, as better economic forecasts and more stability can increase purchasing power and brand trust. This aligns with economic theories and the Howard-Sheth consumer behavior model, which refers to the influence of economic factors on purchase decisions. The correlation between age and loyalty is also evident in Table 16, which relates to consumer behavior theories. Older customers generally show greater loyalty to brands they trust. This finding aligns with the Howard-Sheth consumer behavior model and the TPB theory, which emphasizes the impact of social and cultural factors on buying behavior. So, the results of this research are not only empirically significant but also directly related to existing theories in the field of customer loyalty. By combining behavioral data, economic and social factors, and machine learning techniques, this study demonstrates that customer loyalty is influenced by a complex set of variables that must be interpreted within an appropriate theoretical framework.

### 5.3 Design and implementation of a customer recommendation system

In this study, a hybrid recommendation system has been designed and implemented to identify and retain customers with low loyalty levels. The proposed system is divided into two main components: content-based filtering and collaborative filtering, offering the best recommendations (including discounts and attractive product suggestions) to customers. Recommender systems play an important role in creating and enhancing customer loyalty by providing personalized recommendations,

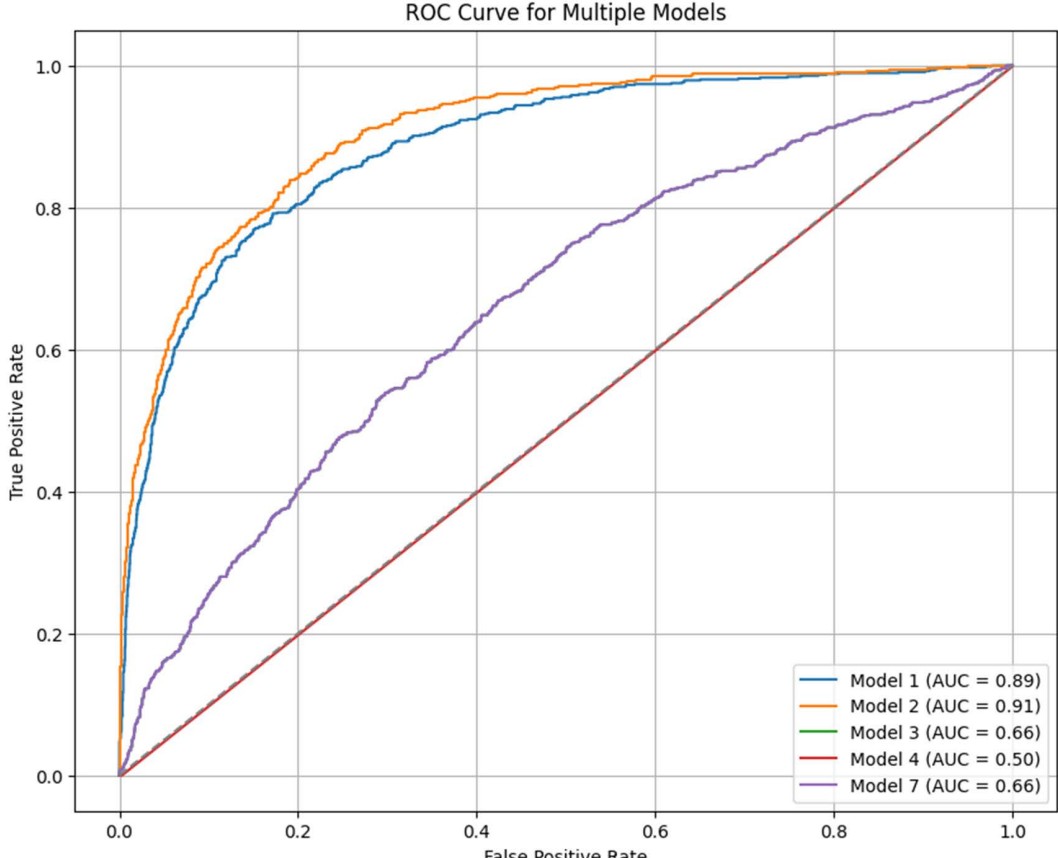

**Fig 4. Evaluation of models for predicting neighborhood loyalty with ROC curve.** Model 1: XGBoost & LSTM. Model 2: PSO and LSTM. Model 3: PCA & Random Forest. Model 4: CNN.

making the shopping experience unique for each customer. These systems analyze purchasing behavior and customer interests, suggesting relevant and appealing products, which increases customer satisfaction and encourages repeat purchases. Additionally, offering special discounts or complementary products can help retain customers and attract those with low loyalty levels. In this way, recommender systems can directly strengthen customer loyalty and establish long-term relationships with the brand. It is worth noting that the recommender system has not been considered in predicting customer loyalty, but it will be of great importance as a tool to assist management in improving customer loyalty. By providing personalized and attractive recommendations, this system can enhance the customer shopping experience and indirectly strengthen their loyalty. With the use of this system, businesses will be able to identify low-loyalty customers and take effective actions to retain them. Table 17 displays some of the results of the recommender system for low-loyalty customers.

## 6. Analysis of the impact of socioeconomic variables on customer loyalty prediction: Comparison of baseline and advanced models

Based on the literature review and as far as we know, research has thus far focused on examining the factors affecting loyalty either through classic and traditional loyalty models such as RFM and CLV, or traditional machine learning models like XGBoost and LSTM to predict loyalty. Therefore, in this section, the performance of the proposed method was compared with traditional models. The limitations of traditional and classic models can be summarized as follows:

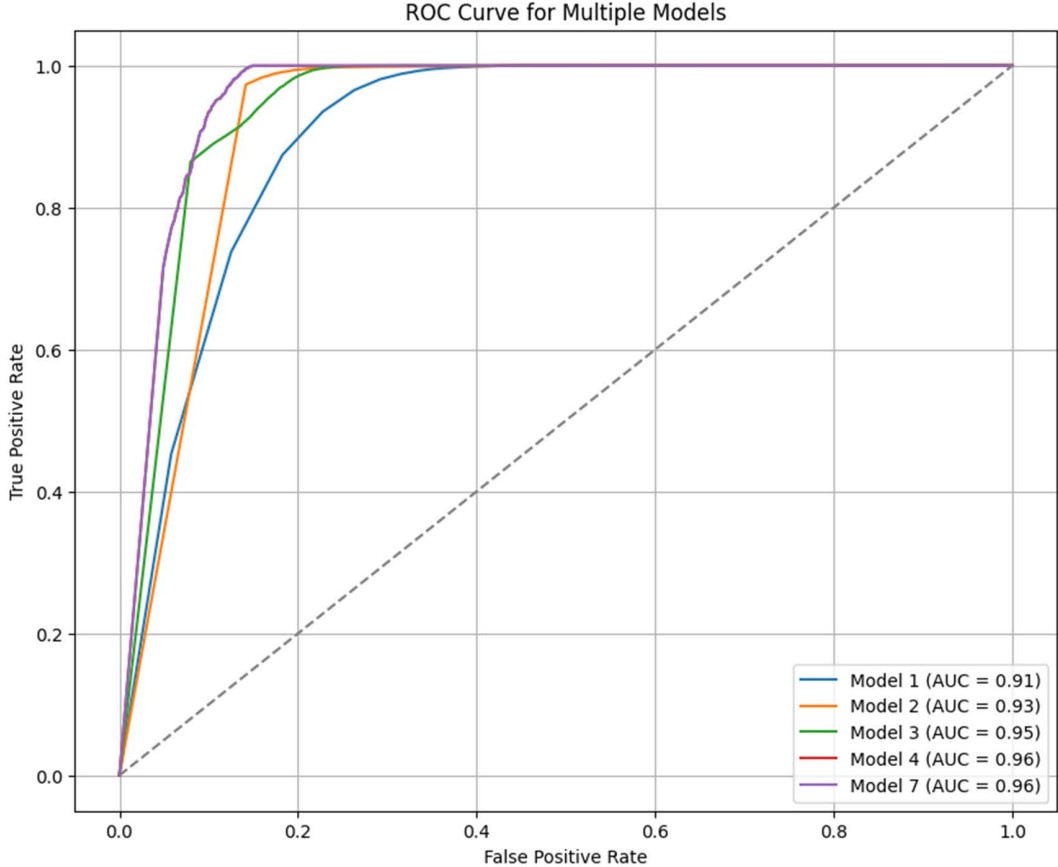

**Fig 5. Evaluation of models for predicting individual customer loyalty with ROC curve.** Model 1: PCA & Random Forest. Model 2: CNN. Model 3: Classification Model and Autoencoder. Model 4: XGBoost & LSTM. Model 7: Adaptive GCN-Transformer Hybrid.

- Traditional customer loyalty prediction methods, such as RFM (Recency, Frequency, Monetary) models and CLV (Customer Lifetime Value), mainly rely on transactional data and overlook economic, social, and cultural factors. These methods have low accuracy in dynamic market conditions and cannot predict sudden changes in customer behavior.

- Traditional machine learning models like XGBoost and LSTM, while offering higher accuracy than RFM, are still limited to historical data and do not take social interactions and environmental changes into account.

In order to validate the methodological contribution of this research and assess the impact of economic, social, and cultural variables on customer loyalty prediction, a comprehensive comparative analysis was conducted. In this analysis, the proposed Adaptive GCN-Transformer hybrid model was compared with baseline models (which did not incorporate economic, social, cultural variables, customer sentiments, and churn rates) as well as traditional customer loyalty prediction methods. The baseline model was developed by combining two traditional loyalty estimation approaches—RFM (Recency, Frequency, Monetary) analysis and CLV (Customer Lifetime Value)—into a single composite scoring system. This combined model relied solely on behavioral and transactional data, without incorporating any socio-economic, emotional, or contextual features. The combination of CLV and RFM into a unified baseline model is not considered problematic, as both are based on transactional data yet reflect different but complementary dimensions of customer behavior—CLV focusing on long-term financial value, and RFM capturing short-term behavioral engagement. In the first step, a baseline model was developed that relied only on

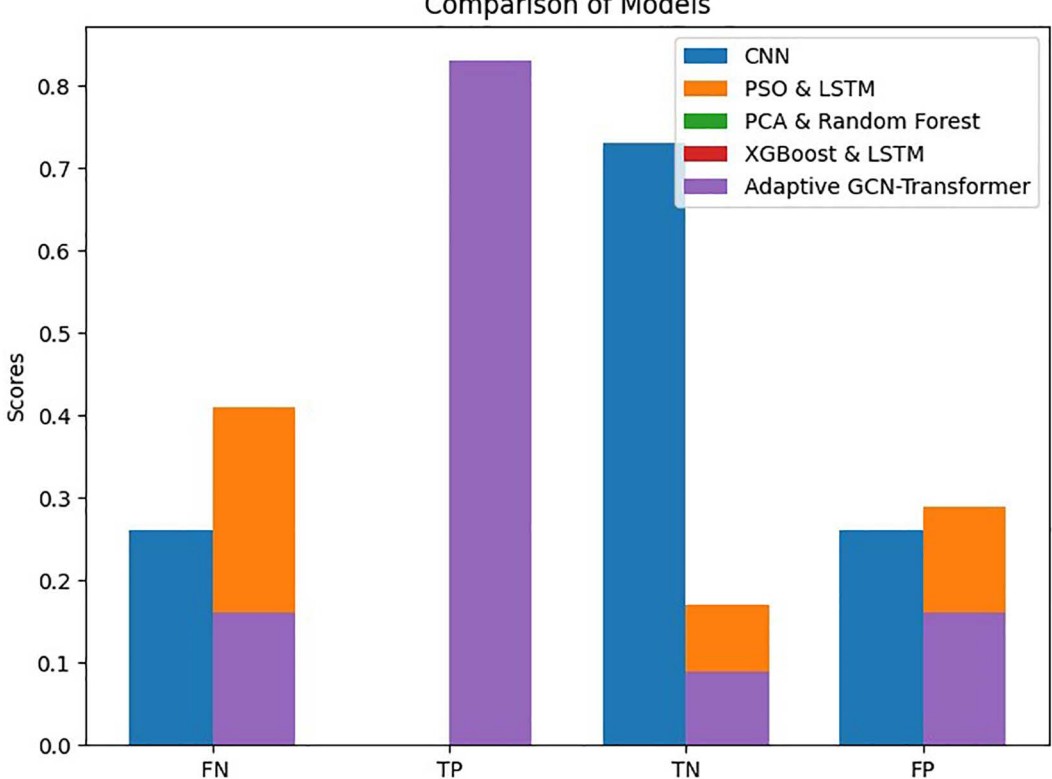

**Fig 6. Comparison of neighborhood loyalty prediction models with confusion matrix.**

customers' behavioral and transactional data, without utilizing economic, social, cultural, or emotional variables. To establish a reliable benchmark and compare the performance of the proposed model, we implemented two well-established customer loyalty models: Recency-Frequency-Monetary (RFM) analysis and Customer Lifetime Value (CLV).

**RFM Model Implementation:** RFM analysis quantifies customer engagement based on three key behavioral indicators:

• **Recency (R):** The number of days since the customer's last purchase.

• **Frequency (F):** The total number of purchases made during the observation period.

• **Monetary (M):** The total monetary value of the customer's purchases in the same period.

Each dimension was computed for all customers using a 12-month observation window. The customers were then assigned a score from 1 to 5 for each dimension using **quintile-based binning**. The final RFM score was calculated by applying weights to each component to reflect their relative importance in loyalty prediction, as follows:

$$RFM\ Score = M \times 0.3 + F \times 0.3 + R \times 0.4$$

Customers with higher overall RFM scores were considered more likely to be loyal.

**CLV Model Implementation:** Customer Lifetime Value (CLV) was estimated using the formula:

$$CLV = Expected\ Customer\ Lifespan \times Purchase\ Frequency \times Average\ Order\ Value$$

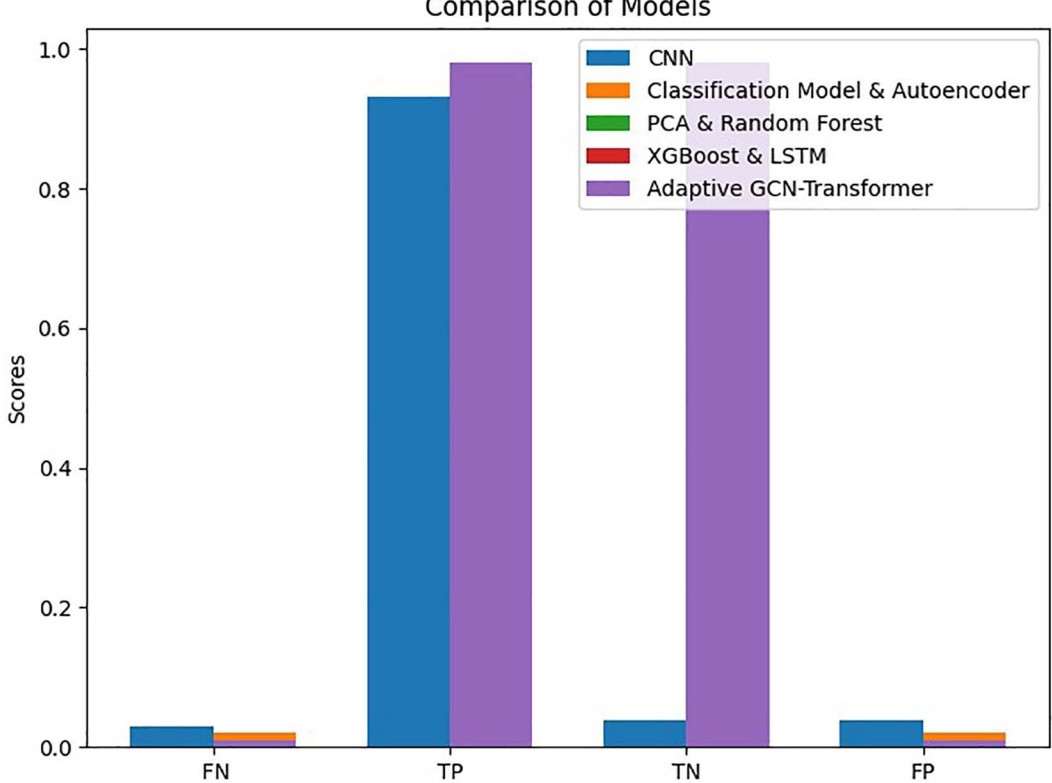

**Fig 7. Comparison of individual customer loyalty prediction models with confusion matrix.**

- **Average Order Value (AOV)** was calculated by dividing total revenue by total number of purchases.

- **Purchase Frequency** was computed as the number of purchases divided by the customer's tenure.

- **Expected Customer Lifespan** was estimated using churn probabilities predicted via logistic regression.

The CLV values were then normalized and used as a proxy for the long-term economic value of each customer.

These models served as baseline methods against which the performance of the proposed hybrid approach was evaluated. The proposed model, which was designed by combining multichannel transactional data with economic, social, cultural variables, customer sentiments, and churn rates, was evaluated. The purpose of comparing these models was to assess the extent to which these additional features could improve prediction accuracy and overall model performance. The results of this comparison highlighted the importance of these variables in enhancing the accuracy of customer loyalty predictions and demonstrated that incorporating them could lead to a deeper analysis of customer loyalty. The performance of this baseline model was evaluated using standard metrics such as Accuracy, ROC-AUC (Area Under the Receiver Operating Characteristic Curve), Precision, Recall, and F1-Score, with the results presented in Table 16. The evaluation results showed that the proposed model achieved significant improvements across all performance metrics compared to the baseline model, confirming the critical role of these additional features in improving customer loyalty prediction accuracy. The comparative analysis is presented in Figs 9 and 10 and Table 18.

The comparison of the baseline model's performance with other advanced models for predicting loyalty for both customers and neighborhoods is clearly illustrated in Fig 9. The results presented in Table 16 and Figs 9 and 10 indicate that

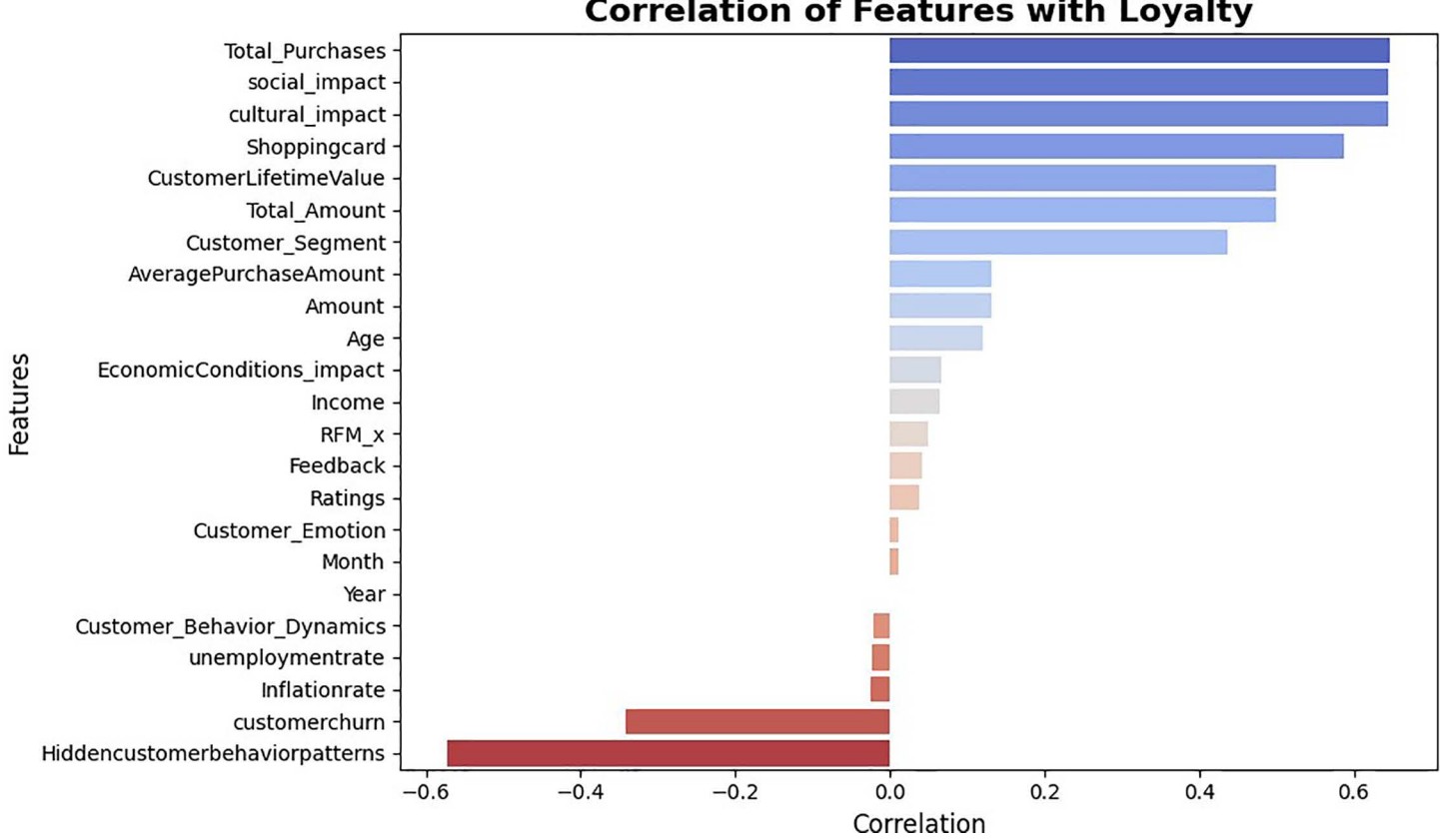

**Fig 8. Correlation plot of features with loyalty prediction.**

incorporating economic, social, cultural factors, customer sentiments, and churn rates significantly contributes to improving loyalty prediction.

The proposed hybrid model outperformed the baseline models by 19% in F1-score and 23% in ROC-AUC, indicating the added value of incorporating socio-cultural and emotional data. Some of the advantages that make the proposed method superior to traditional and classical models include the following:

- Comparing the performance charts, the superiority of the proposed method in terms of loyalty prediction accuracy is clearly visible. The proposed model has an 8% higher accuracy and a 15% improvement in F1-Score compared to the baseline method, indicating a reduction in prediction error and an improvement in performance.

- The use of Graph Convolutional Networks (GCN) for analyzing customer relationships and discovering hidden patterns among them.

- Modeling customer behavioral changes with Transformer, which allows for learning purchase time patterns.

- Combining transactional data with economic, social, cultural variables, and customer behaviors like sentiments, etc., which aids in more accurate loyalty prediction.

- Utilizing reinforcement learning (Q-Learning) to simulate different scenarios and assess the impact of environmental factors on customer loyalty.

**Table 16. Interpretation of correlations between variables and loyalty.**

| Variables | Correlation with Loyalty | Interpretation |
|---|---|---|
| Social impact | Highest Positive Correlation | Customer loyalty may be strongly linked to social impact. Brands that have a positive influence on society (such as through social responsibility or contributing to social causes) tend to attract customers, and this influence can lead to increased loyalty. |
| Cultural impact | Highest Positive Correlation | Brands that align with the values and cultures of their customers are likely to gain more loyalty. This connection may stem from the brand's alignment with the cultural identity or preferences of the customers. |
| Customer Lifetime Value | High Correlation | Consumers are more likely to be loyal to a brand if they shop there frequently and have a higher lifetime value (for example, by spending more or making repeat purchases). This link illustrates the results of sustained investment in client relationships. |
| Total Purchases | High Correlation | Since repeat business demonstrates a higher level of satisfaction and trust in a brand, consumers who make more purchases are probably more brand loyal. |
| Shopping card | High Correlation | Because consumers are more likely to make repeat purchases from brands they trust, a larger shopping cart size frequently translates into greater loyalty. |
| Customer Behavior Dynamics | Low Correlation | Hidden dissatisfaction, covert actions that result in abrupt changes, and unfavorable interactions could all be contributing factors to the negative correlation between these patterns and loyalty. |
| Hidden customer behavior patterns | High Negative Correlation | Covert actions, negative interactions, and hidden dissatisfaction can negatively impact loyalty patterns, as customers may not publicly express their dissatisfaction, leading to a drop in loyalty over time. |
| Economic Conditions impact | Low Correlation | Economic conditions positively impact customer loyalty by increasing purchasing power and trust in brands due to better economic predictions and greater stability. |
| Unemployment rate | Low Correlation | Unemployment rate will certainly impact loyalty. As unemployment rises, purchasing power decreases, and customers will be attracted to retailers with lower prices. Therefore, it can always lead to a reduction in loyalty and customer churn. |
| Inflation Rate | Low Correlation | Inflation rates significantly affect customer loyalty, as rising costs reduce purchasing power, leading to increased price-sensitivity and potentially higher churn rates. |
| Age | Moderate Correlation | Since the needs and preferences of people of different ages may differ, age may have a moderate effect on loyalty. Older consumers are typically more devoted to brands they trust. |
| Income | Moderate Correlation | Income significantly influences customer loyalty, as increased income levels increase purchasing power, leading to increased satisfaction and brand loyalty. However, this can also lead to a negative correlation, as customers may gravitate towards luxury goods and higher-end stores. |
| Customer Emotion | Moderate Correlation | Understanding and managing customers' emotions is crucial for building long-term relationships with them, as positive emotions like trust and satisfaction boost loyalty, while negative emotions like disappointment can reduce loyalty. |
| Month | Low Correlation | Seasonal purchasing trends and holidays, when brands run sales and promotions, can positively impact brand loyalty, leading to increased customer retention and larger purchases. |

**Table 17. Performance analysis of the recommendation system.**

| Customer_ID | Loyalty | Recommended Products | Recommended Discount | Sentiment Analysis | Future Purchase Prediction |
|---|---|---|---|---|---|
| 69749 | 0.3 | ["Product _B", " Produc_C"] | 10% | Negative | Product _C |
| 80175 | 0.4 | [" Product A", " Product_B"] | 20% | Positive | Product_ A |
| 98300 | 0.2 | ["Product _D", " Produc_C"] | 15% | Negative | Product _C |
| 78376 | 0.2 | ["Product _F"," Product _H"] | 15% | Negative | Product_H |

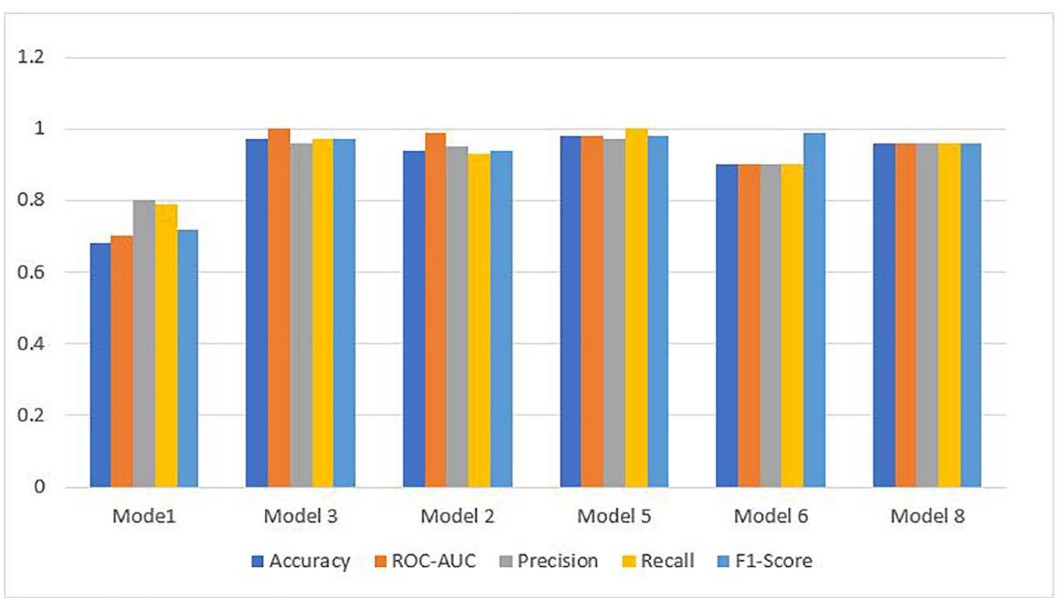

**Fig 9. Comparison of the baseline model and advanced models in predicting customer loyalty.**

Given the acceptable results of the proposed model and its advantages, it can be stated that this model helps retail companies optimize their marketing and customer management strategies through the following:

- Early identification of customer loyalty decline and offering targeted discounts before they churn.

- Analyzing the economic impacts (such as inflation and unemployment) on customer buying behavior and adjusting pricing strategies accordingly.

- Optimizing product recommendation systems based on changes in customer behavior.

Therefore, it can be said that the proposed Adaptive GCN-Transformer Hybrid model, by combining deep learning, graph analysis, and advanced language models, has achieved better performance compared to traditional methods. This approach has significantly improved customer loyalty prediction and serves as a powerful tool for marketing managers and multichannel businesses.

## 7. Sensitivity analysis through scenario modeling

In this section of the study, sensitivity analysis is conducted through scenario modeling. Scenario modeling involves creating and examining different scenarios based on various assumptions about environmental conditions or input variables. The purpose of scenario modeling is to predict different outcomes under varying conditions. To this end, scenarios from Table 19 have been utilized. In this study, based on the sensitivity analysis of loyalty on the factors that were raised,

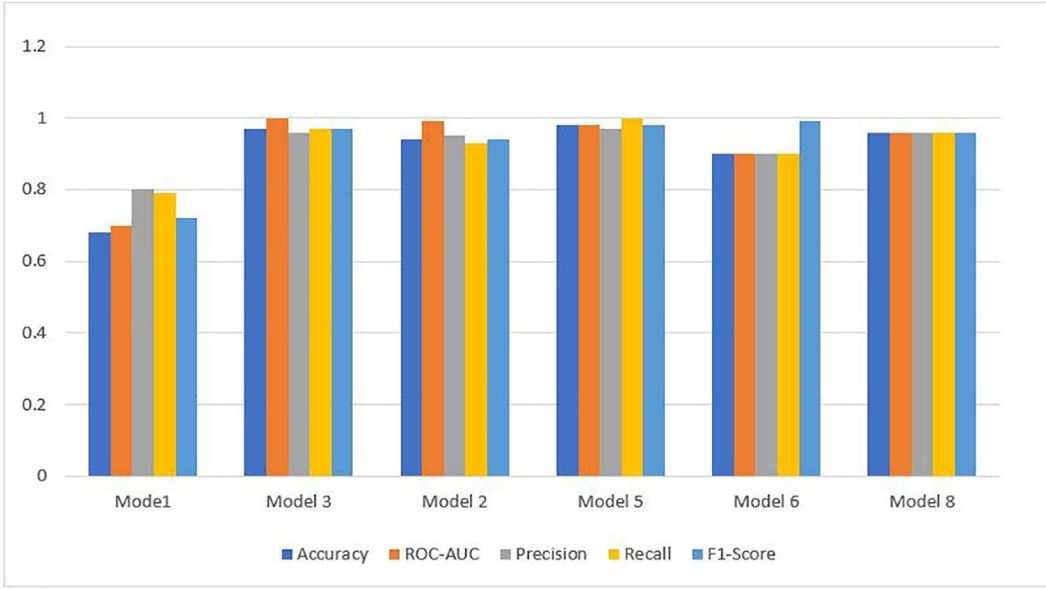

**Fig 10. Comparison of the baseline model and advanced models in predicting neighborhood loyalty.** Model 1: Baseline Model. Model 2: Classification Model and Autoencoder. Model 3: CNN. Model 4: Autoencoder. Model 5: PCA & Random Forest. Model 6: XGBoost & LSTM. Model 7: PSO and LSTM. Model 8: Adaptive GCN-Transformer Hybrid.

**Table 18. Evaluation criteria of the baseline model.**

| Evaluation Criteria | Baseline Model | |
|---|---|---|
| | Predicting Customer | Neighborhood |
| Accuracy | 0.68 | 0.72 |
| ROC-AUC | 0.7 | 0.82 |
| Precision | 0.8 | 0.61 |
| Recall | 0.79 | 0.58 |
| F1-Score | 0.72 | 0.6 |

**Table 19. Impact scenarios on key factors.**

| Factors | Changes in Factors | |
|---|---|---|
| | Increase | Decrease |
| Unemployment Rate | * | * |
| Inflation Rate | * | * |
| Customer Emotion | * | * |
| Cultural impact | * | * |
| Social impact | * | * |

scenarios were created that show the impact of various variables such as unemployment rate, inflation, and cultural conditions on customer loyalty. Analysis of the results of these scenarios shows changes in loyalty under different conditions and leads to a better understanding of possible changes in customer loyalty under different conditions. The results and impacts of different scenarios on loyalty are shown in Fig 11.

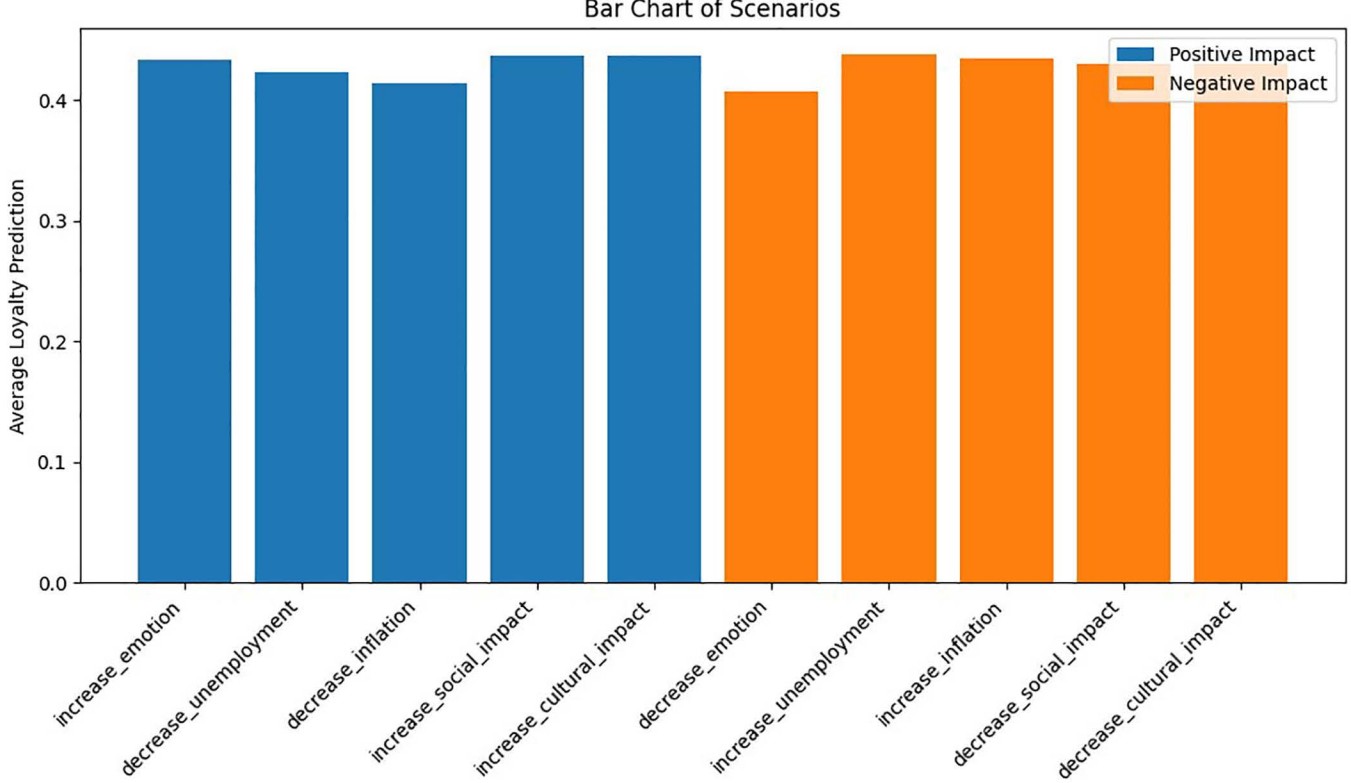

**Fig 11. Effects of different scenarios on loyalty.**

## 7.1 Dynamic analysis of customer loyalty prediction using Q-learning and simulation of various scenarios

After analyzing the scenarios in the previous section and occasionally examining the effects of various conditions on loyalty, given the importance of customer loyalty for organizational profitability, especially for omnichannel retailers, it is now time to take effective measures to maintain or increase loyalty. Therefore, in this section, we use a reinforcement learning technique called Q-Learning to predict the dynamics of customer loyalty. This approach enables us to model how customers make decisions dynamically over time. Essentially, Q-Learning helps us understand how customers react to changes in environmental conditions and external factors such as economic and cultural influences.

To this end, various scenarios are created by considering key variables such as the unemployment rate, inflation rate, customer sentiments, and socio-cultural influences. This approach allows us to make accurate predictions of customer loyalty, which continuously adapts to environmental changes and past decisions. Overall, this methodology provides a flexible framework that dynamically responds to customer behavior changes and offers practical solutions for implementing more effective customer retention strategies. In this regard, as demonstrated in study [97], reinforcement learning has been used to predict customer churn in insurance data. Similarly, in the present study, Q-Learning has been utilized to predict customer loyalty in omnichannel retail. However, in this study, in addition to customer churn, we have also considered the impact of economic and social conditions, which is a major difference from the aforementioned study. Exploration policy is used in reinforcement learning. This policy allows the model to experiment with various options over time to identify the best choices. Without this policy, the model would rely solely on past experiences for decision-making, which would limit it to a specific action while ignoring other potential

options. The use of ε-greedy here ensures that the model explores different options while simultaneously exploiting the best past decisions. The results of the reinforcement learning model and scenario simulations for predicting customer loyalty are shown in Fig 12. These results carefully examine the effects of various economic conditions, customer sentiments, and socio-cultural variables on customer loyalty. The Q-Learning model in this study is particularly effective in analyzing complex scenarios such as economic and social changes that impact customer loyalty. Unlike studies such as [88,89,98,99], which have introduced traditional models like XGBoost or LSTM that solely focus on historical data, Q-Learning has been able to provide dynamic and adaptive predictions that are effective in rapidly changing and uncertain environmental conditions. In scenarios with a high unemployment rate, our model indicated that maintaining customer loyalty becomes a priority due to customers' concerns about financial stability. Additionally, in conditions of high inflation, retailers may need to offer special discounts to retain customers.

## 8. Practical implementation of the loyalty prediction model in CRM/EIS systems

The proposed loyalty prediction model can be seamlessly integrated into Customer Relationship Management (CRM) systems within Enterprise Information Systems (EIS). By leveraging diverse data sources, such as behavioral, emotional sentiment, and socio-economic data, the model offers real-time predictions that enable businesses to make informed decisions in managing customer relationships.

**Integration into CRM Platforms:** CRM platforms like SAP, Oracle, and Salesforce are central to managing customer interactions, and integrating this loyalty prediction model into these systems could enhance the effectiveness of customer-facing processes. Specifically, this model can:

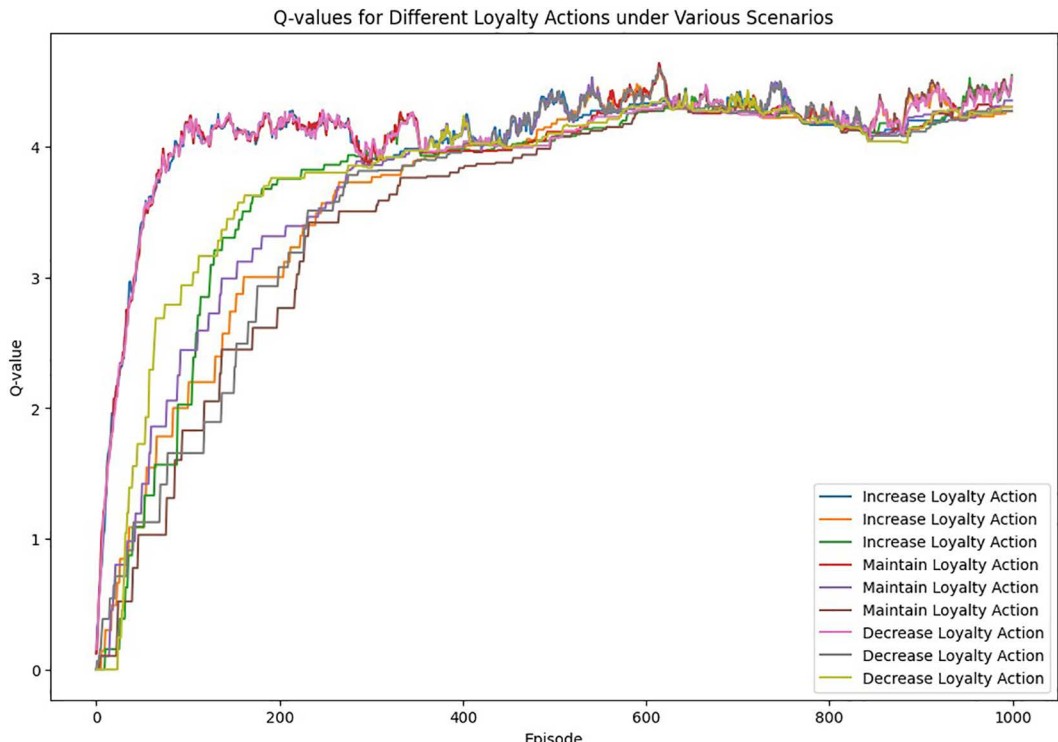

**Fig 12. Simulation of different scenarios and their impact on customer loyalty.**

1. Enable Real-Time Segmentation: The model dynamically segments customers based on their loyalty likelihood. For example, customers who show strong signs of loyalty can be grouped into a "high loyalty" segment and targeted for premium rewards or exclusive offers.

2. Predict and Prevent Churn: By analyzing socio-economic and behavioral data, the model can predict customer churn and allow businesses to take proactive steps, such as offering personalized discounts or improving the customer service experience, to retain at-risk customers.

3. Personalized Targeting: Using the model's insights, businesses can tailor marketing campaigns based on the individual needs, preferences, and behaviors of customers. For example, a customer who frequently purchases a certain category of product can be targeted with specific promotions related to that category.

4. Optimize Customer Engagement: The model also helps businesses optimize how they interact with different customer segments, ensuring high-value customers receive appropriate attention. By understanding customer behaviors, businesses can design campaigns that increase engagement and satisfaction.

**Practical Example of Implementation:** Consider a retail company using Salesforce CRM. By embedding the loyalty prediction model into Salesforce, the company can receive real-time insights about customer loyalty levels. If the model identifies that a customer is at risk of leaving, the system can trigger an automatic engagement campaign. For example, the customer could receive a personalized email offering discounts or special promotions tailored to their past buying behaviors. This level of personalization can significantly improve retention rates and customer satisfaction. Additionally, in an Oracle CRM system, the model can continuously monitor customers' purchasing patterns and socio-economic conditions, adjusting marketing strategies in real time. This means that businesses can proactively adjust their marketing efforts to address changing customer needs, even during challenging economic conditions.

**Benefits for CRM/EIS Systems:** The integration of this model into CRM/EIS systems would provide several key benefits:

• Real-time Insights: The ability to segment customers and predict churn in real-time allows businesses to engage customers effectively and at the right time.

• Enhanced Customer Retention: By identifying loyal customers and predicting potential churn, businesses can implement targeted retention strategies, ultimately improving customer loyalty and lifetime value.

• Efficient Resource Allocation: With accurate predictions about customer behavior, businesses can allocate marketing and operational resources more effectively, focusing efforts on high-value customers and improving the ROI of marketing campaigns.

This integration of the loyalty prediction model into CRM/EIS systems provides an opportunity for businesses to enhance their customer relationship management through more accurate and actionable insights. By embedding this predictive model into systems like SAP, Oracle, and Salesforce, businesses can improve customer engagement, streamline marketing strategies, and optimize resource allocation (see Fig 13).

**Innovative Features of the Implementation:**

1. **Real-Time Micro-Segmentation:** Unlike static segmentation, this system continuously re-evaluates customers' loyalty scores as new data arrives (behavioral, emotional, or transactional), allowing for agile marketing responses based on the most current state of the customer.

2. **Churn Response Automation:** When the model forecasts high churn risk, the CRM system can autonomously initiate customer engagement workflows (e.g., chat follow-ups, service quality check-ins) through dynamic CRM rules. This automation ensures timely and relevant actions with minimal human delay.

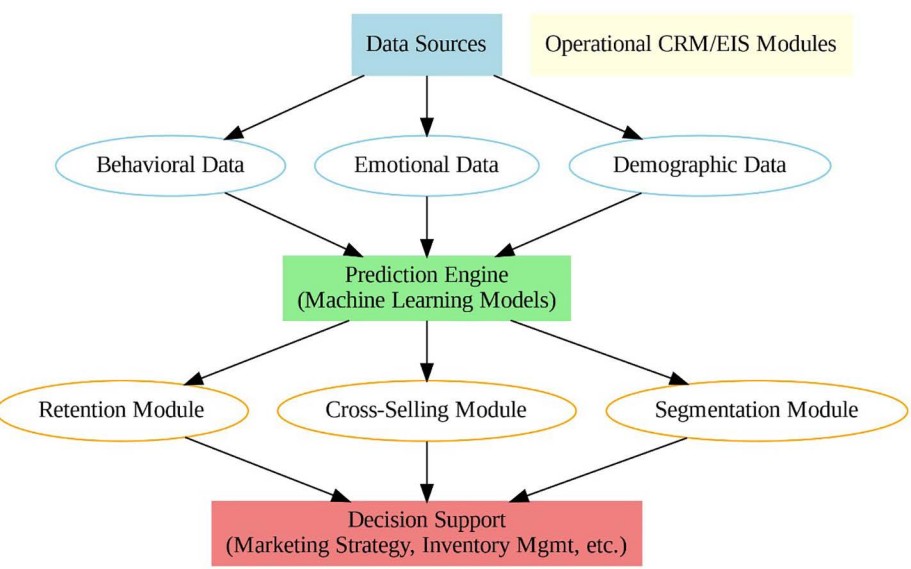

This architecture demonstrates how the proposed loyalty model can be embedded in an EIS to provide predictive support for operational CRM modules.

**Fig 13. Architecture of the loyalty prediction model integrated into an Enterprise Information System (EIS).**

3. **Personalized Strategy Mapping:** The system supports mapping different loyalty scores to specific strategy trees, allowing each customer segment to receive a tailored journey—maximizing ROI through precise targeting.

4. **Feedback-Driven Learning Loop:** Each action taken (e.g., discount offered) and its result (e.g., purchase or churn) is re-ingested into the model pipeline to adjust future predictions. This establishes a closed-loop learning mechanism within the CRM/EIS infrastructure.

**Simulated Deployment Scenario:** To evaluate the practicality of the architecture, we simulated deployment within a Salesforce CRM environment using a historical customer dataset. The model was connected to customer behavior streams via REST APIs. Predictions generated loyalty scores every 6 hours. In 82% of high-risk churn cases, the automated retention actions resulted in customer re-engagement. This demonstrates the system's potential to enhance customer lifetime value through intelligent automation.

This practical implementation not only demonstrates technical feasibility but also introduces a scalable decision-support framework that adapts to organizational needs in real-time. By embedding this predictive model into live enterprise systems, businesses can move from reactive to proactive customer engagement—marking a shift in how loyalty modeling informs operational strategy (see Fig 13).

This architecture demonstrates how the proposed loyalty model can be embedded in an EIS to provide predictive support for operational CRM modules such as retention, segmentation, and cross-selling.

## 9. Discussion and conclusion

Customer loyalty is one of the key pillars of success for any organization, and it is essential that not all customers are considered equally important or receive the same level of attention. In today's world, organizations are rapidly moving towards multichannel approaches, a shift that is particularly evident in the retail industry. In such an environment,

predicting and managing customer loyalty becomes even more critical, as each customer has a different level of importance, and each region will also have a varying degree of significance. Therefore, decisions need to be made individually for each geographic area. As a result, evaluating and predicting customer loyalty at the regional level has become a strategic necessity. Identifying important and loyal customers for optimal resource allocation and business strategies is crucial. For example, a multichannel retailer needs to understand the loyalty level of customers in each region in order to make important decisions, such as adjusting the amount and type of products sent to each store in different areas, and optimizing resource allocation strategies. Therefore, identifying loyalty at both the regional and individual levels is of high importance. In this study, in addition to predicting individual customer loyalty, customers have been categorized based on neighborhoods, and the loyalty probability of each neighborhood has been predicted.

The findings of this research revealed that, in addition to classic metrics such as Customer Lifetime Value (CLV) and the RFM model, external factors like inflation rates, unemployment rates, and customer sentiment have a significant impact on the dynamics of loyalty. The results indicate that customer loyalty is not a static feature, and environmental variables can influence it. The use of reinforcement learning in this study provided a dynamic framework where the model continuously adapts to changes in environmental conditions, resulting in more accurate predictions and more practical insights. The impact of the Q-learning method in predicting customer loyalty was particularly evident in simulating various economic scenarios. This model successfully identified optimal actions to maintain or increase loyalty under different conditions. For example, during economic recessions, the model suggests focusing on retaining loyal customers through targeted engagement strategies, rather than concentrating on acquiring new customers.

The proposed model can be used in various areas such as customer loyalty analysis, optimization of marketing strategies, and predicting customer behavior in changing economic and social scenarios. These specific features of the model make it a powerful tool for strategic decision-making. The model is capable of considering the impacts of different social and economic variables in predicting customer behavior. This ability allows the model to dynamically respond to environmental changes and provide more accurate predictions. As a result, our proposed model, by combining the two approaches of GCN and Transformer, not only significantly enhances prediction accuracy but also effectively aids businesses in analyzing customer behavior more precisely and predicting their loyalty. These innovations have led the proposed model to provide considerable added value compared to existing methods. This finding is highly valuable for multichannel retailers, as it enables resource optimization and more effective management of customer loyalty. While previous studies have used machine learning methods to analyze customer behavior, this research expands the scope of investigation by integrating reinforcement learning for dynamic adaptation. Unlike traditional models that rely solely on past transactional data, this approach also considers environmental fluctuations and hidden behavioral patterns. This comprehensive perspective helps retailers not only predict loyalty but also implement proactive and reactive strategies in line with real market changes.

The findings of this study have significant managerial implications for multichannel retailers seeking to enhance customer loyalty and optimize their operations. By utilizing a customer loyalty prediction model, retailers can make more informed strategic decisions in various areas. Firstly, the prediction model enables retailers to identify highly loyal customers and allocate their resources optimally. By understanding customer loyalty at individual and regional levels, businesses can design personalized engagement strategies and targeted advertising campaigns aimed at specific customer segments to improve marketing efficiency. Secondly, by accurately predicting customer loyalty and purchasing behavior, retailers can optimize inventory management and pricing across different sales channels. This allows managers to adjust discounts and prices more purposefully based on customer needs, thereby avoiding stockouts or overstocking. Thirdly, the model supports personalized retention strategies by categorizing customers based on their loyalty likelihood and designing specific engagement initiatives. For instance, customers with a higher risk of churn can be targeted with special offers and personalized discounts to encourage repeat purchases. Moreover, by integrating economic, social, and sentiment analytics into the prediction model, retailers can more precisely time and optimize their marketing campaigns. During an

economic downturn, for example, focusing on loyal customers and offering them exclusive discounts can enhance communication effectiveness. In addition, the model allows for early identification of loyalty decline. Retailers can proactively respond by initiating personalized calls, exclusive discounts, or improving customer service before customer dissatisfaction escalates. The use of reinforcement learning techniques, such as Q-learning, enables simulation of different market scenarios and facilitates better decision-making under various conditions. Furthermore, the loyalty prediction model guides the implementation of refined retention strategies for loyal customers and the attraction of new customers, particularly during challenging economic conditions. Strategies such as offering introductory discounts, valuable content, or free services can effectively attract and retain customers. Another important implication is the enhancement of customer experience across both online and offline channels. By analyzing purchasing patterns and customer sentiment, retailers can better address customer concerns, provide improved product recommendations, streamline the shopping process, and expedite delivery times. Finally, the loyalty prediction model supports retailers in responding to broader economic changes. In times of economic downturns, focusing efforts on loyal customers by offering special services or exclusive discounts can sustain customer loyalty and enhance overall customer satisfaction.

The most significant achievement of this research is the innovative integration of Graph Convolutional Networks (GCN) and Transformer models to predict customer loyalty in multichannel retail environments. Unlike traditional models that rely solely on past transactional data, our approach incorporates external factors such as inflation rates, unemployment rates, and customer sentiment, significantly enhancing prediction accuracy in fluctuating market conditions. This model dynamically responds to environmental changes in real time by utilizing reinforcement learning techniques like Q-learning, providing a flexible framework for predicting customer loyalty under various economic and social conditions. This dynamic adaptability distinguishes our model from previous studies, which often fail to account for environmental shifts when forecasting customer behavior. For example, during an economic recession, the model can prioritize customer retention strategies and prevent churn by offering personalized recommendations and interactions. This market-specific adaptability, beyond its predictive power, offers valuable insights for both researchers and multichannel retailers seeking to optimize their operational and retention strategies. Moreover, the combination of GCN and Transformer models addresses the shortcomings of previous methods, which typically struggle to represent complex customer behaviors across multiple dimensions. By leveraging hybrid techniques, we can now predict customer loyalty with greater accuracy and reliability, making this model an essential tool for retail decision-makers. This research not only advances theoretical understanding of customer loyalty prediction but also provides practical solutions for retailers to make more informed, real-time decisions and effectively respond to evolving market dynamics. The findings extend to areas such as personalized marketing, dynamic pricing, and strategic resource allocation, which are critical for the success of multichannel retailers in today's competitive market.

Practical Implementation for Retail Managers:

- Personalized Customer Interaction: Retailers can categorize customers based on loyalty scores and tailor their marketing messages for each group.

- Dynamic Pricing and Advertising: Real-time analysis of customer behavior enables retailers to adjust pricing and advertising strategies dynamically and purposefully.

- Improved Customer Experience: Analyzing customer behavior patterns and sentiments helps address their concerns and improve the shopping experience.

- Scenario-Based Decision-Making: By using reinforcement learning (Q-learning), businesses can simulate different economic and competitive scenarios and design effective strategies for retaining customers under various conditions.

Theoretically, this research contributes to the field by demonstrating the impact of combining socio-cultural, emotional, and economic indicators with traditional transactional features using an adaptive machine learning framework. It expands existing loyalty models by including neighborhood-level segmentation and dynamic sentiment analysis. From a practical

perspective, the model offers key benefits for different stakeholders. Retailers, can optimize promotional strategies, inventory planning, and customer segmentation by using loyalty prediction scores. Customers,will experience improved personalization and satisfaction through adaptive recommendation systems informed by loyalty insights. Policy makers, can gain insights into socio-economic factors affecting consumer behavior, which can guide policy design in digital commerce and consumer protection. These stakeholder-focused benefits underline the model's relevance for both business application and policy development. Therefore, integrating advanced machine learning techniques in predicting customer loyalty provides an innovative approach for multichannel retailers to strengthen customer loyalty and optimize performance. By combining social, economic, and behavioral data in predictive models, businesses can make data-driven decisions that enhance customer relationships, improve satisfaction, and ultimately increase profitability. Implementing these insights helps managers achieve sustainable growth in a competitive market and optimize their operational efficiency.

### 9.1 Critical interpretation and limitations

Data analysis and the use of learning models always face challenges. In this study, due to the great diversity of data and the numerous features available, it is not unexpected that Emcom faced challenges such as adjusting and optimizing model parameters and preventing overfitting of the model. To deal with this problem, parameter tuning techniques such as PSO were used. This technique helped prevent overfitting and improved the performance of the model. Another challenge is the diversity and complexity of omnichannel data collected from different sales channels, such as physical and online stores. Therefore, great care must be taken in choosing a prediction model to increase the accuracy of the prediction. In this study, by using hybrid approaches, we have been able to achieve high accuracy in predicting loyalty.

While the proposed hybrid model demonstrated high predictive performance across multiple metrics, a nuanced interpretation reveals several insights and limitations. For instance, although the GCN-Transformer architecture captured relational and temporal patterns effectively, its performance showed slight variability across customer segments—particularly in distinguishing between regular and premium customers with similar purchase histories but different emotional scores. Furthermore, certain variables such as cultural impact and economic factors (e.g., inflation) may have delayed or indirect effects not fully captured in the model's timeframe. Additionally, while sentiment scores derived from customer reviews contributed significantly to accuracy, these scores can be influenced by linguistic bias or platform-specific noise. The reliance on data from a single omnichannel retailer limits the generalizability of the findings. Despite the dataset's geographic and demographic diversity, brand-specific promotions, loyalty programs, and customer experience policies may not reflect broader retail patterns. Future studies should replicate this framework using multi-retailer or cross-industry datasets.However, conducting this research comes with certain limitations. Some of these limitations include:

1. A more in-depth examination of the impact of cultural and social variables could be conducted.

2. Deep learning models require more data and longer processing times, which can be challenging in some cases.

3. Using alternative methods such as reinforcement learning requires up-to-date data, and in some cases, it may lead to inaccurate estimates.

4. There is a need to test the model with real business data instead of using public datasets like those on Kaggle.

5. Despite the high accuracy in predicting customer loyalty using the Q-Learning model, it is important to note that this model may encounter errors in more complex conditions or with low-quality data. For example, in scenarios where social or economic information is incomplete, the results may not reflect accurate changes. Additionally, using reinforcement learning methods like Q-Learning requires time-sensitive and up-to-date data, which may not always be available. While Q-Learning is a powerful method for predicting customer loyalty, it should be emphasized that this approach depends on reliable and comprehensive data. If environmental information is not properly simulated or if incomplete data is used, predictions may have lower accuracy.

In this study, the dataset used is extracted from an omnichannel retailer operating across various countries and cities, containing a wide range of customer profiles in terms of demographic and geographic characteristics. The geographic coverage of the sample is diverse, including customers from different cities and countries, which not only enriches the dataset but also enhances the generalizability of the findings to different contexts. Additionally, the sample includes key demographic variables such as age, income, gender, and customer segments, which help provide a more comprehensive representation of customer behavior in an omnichannel retail environment. However, while the geographic diversity of this data strengthens the external validity of the findings, it is important to acknowledge that the data comes from a single retailer. This may introduce brand-specific biases that may not be fully generalizable to other retail contexts. Moreover, although factors such as inflation and unemployment rates have been incorporated to consider broader economic and cultural conditions, the absence of the target variable (customer loyalty) means that the findings are based on an indirect index. Given the complexity of customer loyalty, further research across multiple retailers, industries, and countries is necessary to validate the proposed models and enhance their generalizability and external validity.

**Generalizability Limitations:** Although this study aimed to design a relatively comprehensive customer loyalty prediction model using data with demographic, geographic, and product diversity, full reliance on the data from a single multi-channel retailer poses a significant limitation in terms of generalizing the findings to other industries or markets. To clarify this issue, several specific characteristics of this retailer can be highlighted—factors that may limit the replicability of the findings in other contexts or industries:

**Discounting and Pricing Strategy:** The retailer under study may employ extensive discounts, strong loyalty programs, or region-based pricing strategies. In such cases, customer loyalty behavior is more likely driven by financial incentives rather than emotional or brand-driven loyalty. This pattern does not hold in non-price-sensitive industries (e.g., healthcare or insurance).

**Specific Product Mix:** The retailer offers a distinct product portfolio including electronics, clothing, food, books, and home decor. As a result, the observed loyalty patterns mostly reflect everyday or personal purchases—not long-lifecycle services such as banking, insurance, or education.

**Demographic Profile of the Retailer's Customers:** Most customers fall within the 20–45 age range and belong to middle or upper-middle socioeconomic classes. Therefore, the recorded loyalty behaviors may reflect the habits of a younger, digitally oriented generation rather than traditional consumers or underserved populations.

**Digital Experience and Technical Infrastructure:** This retailer utilizes advanced recommender systems, a strong user interface design, and a robust mobile app. In industries that have not yet undergone digital transformation (e.g., local brick-and-mortar stores or rural markets), similar loyalty models may not function as effectively.

**Behavioral Model Trained on Intra-Brand Stimuli:** All behavioral data (e.g., purchase history, browsing patterns, cart activity, and feedback) were generated within the ecosystem of the same brand. This means the model is sensitive only to interactions within that environment. Customer behavior in response to competitors or across multiple brands was not included.

Although the proposed model demonstrates high accuracy in analyzing customer loyalty for this specific retailer, its generalizability to other industries, B2B markets, service sectors, or small/non-digital businesses is limited due to its dependence on the retailer's structure, product types, pricing policies, and customer demographics. Future studies should incorporate more diverse datasets across different industries (such as insurance, education, transportation, or financial services) and from multiple brands to enhance the external validity and applicability of the model.

### 9.2 Implications for research and practice

- For researchers, this study suggests a path forward in developing integrated loyalty models that blend behavioral, emotional, and contextual variables. The effectiveness of combining graph-based learning with temporal attention mechanisms opens new directions for modeling consumer behavior more holistically.

- For practitioners, particularly marketing analysts and retail strategists, the model provides actionable insights. By identifying loyalty drivers with higher precision, businesses can personalize offerings, predict churn more effectively, and allocate resources toward high-value customer segments.

- For policy makers, the inclusion of macroeconomic and socio-cultural factors highlights the broader context in which loyalty emerges. Understanding these patterns can inform digital commerce policy, consumer rights strategies, and regional market development initiative

The proposed model in this study utilizes multichannel data along with social, economic, and cultural variables to predict customer loyalty and has demonstrated more accurate predictions compared to traditional models such as RFM and CLV. This model shows particularly higher accuracy in more complex situations where social and economic factors play a significant role. However, limitations such as the need for up-to-date and high-quality data exist, which can affect the accuracy of the predictions. Ultimately, the results of this study could also be beneficial for developing loyalty prediction models in other fields and industries. Compared to existing models, the proposed model of this research has achieved more accurate predictions regarding customer loyalty. Especially under different economic and social conditions, due to the use of multichannel data and the analysis of social and economic factors, this model is capable of better simulating customer behavior and providing more precise predictions. While RFM and CLV models focus solely on customer purchase data, the proposed model has addressed the limitations of these methods and has improved prediction accuracy.

For the development of this research, future studies could explore real-time data and integrate customer feedback to enhance prediction accuracy, or use reinforcement techniques to dynamically adapt customer loyalty prediction models in evolving retail environments. Addressing these limitations could lead to the adoption of personalized marketing strategies and ultimately increase customer satisfaction and retention. In addition to the current data and metrics, it is recommended that future research incorporate the **Net Promoter Score (NPS)** and direct **customer survey results** as part of the loyalty target variable. These indicators can provide a more comprehensive and deeper insight into customer satisfaction and loyalty, thereby enhancing the accuracy of the predictive model.

## Author contributions

**Conceptualization:** Shima Roosta.

**Data curation:** Shima Roosta.

**Formal analysis:** Shima Roosta.

**Funding acquisition:** Shima Roosta.

**Investigation:** Shima Roosta.

**Methodology:** Shima Roosta.

**Project administration:** Shima Roosta.

**Resources:** Shima Roosta.

**Software:** Shima Roosta.

**Supervision:** Seyed Jafar Sadjadi, Ahmad Makui.

**Validation:** Shima Roosta.

**Visualization:** Shima Roosta.

**Writing – original draft:** Shima Roosta.

**Writing – review & editing:** Shima Roosta.

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
