## [Decision Letter · Decision Letter 0]

16 Jul 2025

Dear Dr. Roosta,

Thank you for submitting your manuscript to PLOS ONE. After careful consideration, we feel that it has merit but does not fully meet PLOS ONE’s publication criteria as it currently stands. Therefore, we invite you to submit a revised version of the manuscript that addresses the points raised during the review process.

We look forward to receiving your revised manuscript.

Kind regards,

Vincenzo Basile, PhD

Academic Editor

PLOS ONE

2. In the online submission form, you indicated that [The dataset used in this study is available on Kaggle at the following link: [https://www.kaggle.com/datasets/bhavikjikadara/retail-transactional-dataset?resource=download]. The data can be accessed upon reasonable request. If any ethical, privacy, or security concerns arise, access to the data may be restricted accordingly].

Reviewers' comments:

Reviewer's Responses to Questions

**Comments to the Author**

1. Is the manuscript technically sound, and do the data support the conclusions?

Reviewer #1: Yes

Reviewer #2: Partly

2. Has the statistical analysis been performed appropriately and rigorously?

Reviewer #1: Yes

Reviewer #2: No

3. Have the authors made all data underlying the findings in their manuscript fully available?

Reviewer #1: Yes

Reviewer #2: No

4. Is the manuscript presented in an intelligible fashion and written in standard English?

Reviewer #1: Yes

Reviewer #2: No

Reviewer #1: The paper entitled “Predicting Customer Loyalty in Omnichannel Retailing Using Purchase Behavior, Socio-Cultural Factors, and Learning Techniques” is interesting and apply a few models based on modelling and prediction.

The topic is challenging in this era of relationship based on using AI and modelling, but in order to be publisehd the authors need to make some improvements:

-In the Introduction, the authors need to add text about the novelty of this paper, and the gap in the literature review and empirical comparing to their own study;

-To Literature review the authors need to add some updated sources for Sentiment analysis, for BERT model, Evaluation Metrics and also for Hyperparameter Tuning, there are no sources.

-The Methodology is well done and explained, but needs some improvements such as: for the regression equation determined by the authors must be added a clear explanation of the increasing/decreasing of the values for each independent variable and their influences on dependent variable how is explained (loyalty).

-Theoretical implication need to be more developed and the Practical ones, written in italic both, to be easy perceived by the readers, to be more explained for the categories implied: not only for retailers, but also for customers and society/police makers in the field.

Therefore, having these measures for improvements in view, the paper receives major revision.

Reviewer #2: This research paper investigates how to predict customer loyalty in omnichannel retail settings.

1. The paper mentions using a Kaggle dataset, but doesn't fully describe its limitations. A detailed description of the dataset's characteristics (sample size, geographic distribution, temporal coverage, etc.) and a discussion of potential biases is needed. The claim of "diverse demographics, geographic locations, product categories, and brands" needs substantiation with specific numbers and details. The fact that the data comes from a single, unnamed omnichannel retailer significantly limits the generalizability of the findings. The authors acknowledge this limitation, but a more thorough discussion of the potential biases introduced by this single-source data is needed.

2. The description of data preprocessing is superficial. More detail is needed on how missing values were handled, how categorical variables were encoded, and the specific methods used for feature scaling and normalization. The creation of new variables like "Holiday Status" and the various impact scores requires a more rigorous explanation of the methodology and justification for the chosen methods. The lack of detail makes it difficult to assess the validity and reliability of these engineered features.

3. The authors acknowledge the limitations of generalizing findings to different markets or industries, but a more in-depth discussion of the specific factors that might limit generalizability is needed. The reliance on a single retailer's data significantly impacts the external validity of the study.

4. While the paper mentions using several machine learning models, the rationale behind choosing specific models (BERT, Reinforcement Learning, GCN, Transformer) and their combinations needs further justification. The authors should clearly articulate why these specific models are appropriate for this problem and how they address the limitations of previous research. A more detailed comparison of the strengths and weaknesses of each model in relation to the research question is necessary.

5. The paper lacks detail on hyperparameter tuning for the various models. How were the hyperparameters selected and optimized? Were techniques like cross-validation used? This information is crucial for assessing the reproducibility and reliability of the results.

6. The paper uses several evaluation metrics (Accuracy, ROC-AUC, Precision, Recall, F1-score), but a more thorough discussion of the choice of these metrics and their interpretation in the context of the problem is needed. Why were these specific metrics chosen, and what are their limitations? The authors should also discuss the potential for class imbalance in the dataset and how this was addressed.

7. The comparison with baseline models (RFM and CLV) is insufficient. The paper needs to provide a more detailed description of the baseline models used and how they were implemented. A more rigorous comparison of the proposed model's performance against these baseline models is necessary to demonstrate the added value of the proposed approach.

8. The use of SEM to determine weights for combining different indicators into a loyalty score requires more explanation. The authors should provide more detail on the SEM model specification, the model fit indices, and the justification for the chosen weighting scheme. The interpretation of the SEM results needs to be more thorough and nuanced.

9. The paper needs to provide more detail on the experimental setup, including the software and hardware used, to ensure the reproducibility of the results. The code or a detailed description of the implementation should be made available.

10. The interpretation of the results needs to be more critical and nuanced. The authors should discuss the limitations of the findings and potential sources of error. The discussion should also address the implications of the findings for both researchers and practitioners.

**Do you want your identity to be public for this peer review?** For information about this choice, including consent withdrawal, please see our Privacy Policy

Reviewer #1: **Yes: ** Assoc. Prof. PhD Habil. Florea Nicoleta Valentina

Reviewer #2: **Yes: ** Dr. Rinku Sharma Dixit

---

## [Author Response · Author response to Decision Letter 1]

20 Jul 2025

Title: Response to the Reviewers' Comments

We would like to express our sincere gratitude to the esteemed reviewers for their valuable and constructive feedback. All comments have been thoroughly reviewed, and the necessary changes have been made to the article. Here, we present our responses to the reviewers' comments in detail. We hope that we have understood the valuable comments of the esteemed reviewers and have made the appropriate changes.

Reviewer 1:

We sincerely thank you for the valuable suggestions you provided, which have significantly contributed to improving our manuscript. We are truly grateful for the time and effort you dedicated to reviewing our work.

1-In the Introduction, the authors need to add text about the novelty of this paper, and the gap in the literature review and empirical comparing to their own study

Answer: Thank you for your valuable feedback. The requested item has been added to the introduction as follows:

Despite numerous studies on customer loyalty in multi-channel retail environments, significant gaps still exist in this area. Most previous research has focused on analyzing sales data and online customer behavior, addressing only limited factors such as purchase history and transaction data. However, a more accurate understanding of customer behavior requires analyzing multi-channel data and examining social, economic, and cultural variables that significantly influence purchasing decisions and customer loyalty. Additionally, many studies have been limited to transactional data and online customer behavior, with the impact of factors such as economic and cultural conditions on customer loyalty not being thoroughly investigated. Advanced machine learning models, such as Graph Convolutional Networks (GCNs) and Transformer models, can simulate complex and nonlinear relationships in data and provide more accurate predictions. However, the use of these models for predicting customer loyalty has not yet been extensively explored. Many existing studies on loyalty have focused primarily on loyalty drivers or predicting loyalty using sales data and online behavior analysis, often concentrating on one or two sales channels (such as physical stores or online platforms). In contrast, customer interactions with retailers occur across multiple channels, including websites, physical stores, and social media platforms. To address these research gaps, this study aims to develop and expand loyalty prediction models by considering not only transactional data but also social, cultural, economic, and emotional factors, RFM analysis, Customer Lifetime Value (CLV), churn rate, and customer behavior analysis—including shopping cart analysis, customer behavior dynamics, latent behavioral patterns, and sales data—to achieve more accurate predictions of loyalty.

While prior studies have explored customer loyalty using transactional or behavioral data, few have integrated socio-cultural, emotional, and macroeconomic variables into a unified predictive framework. Moreover, traditional models often neglect the dynamic nature of loyalty in omnichannel environments.From an empirical perspective, most previous research has relied on either survey-based data or limited-channel transactional records, often lacking large-scale, real-world data that includes cross-channel behaviors. In contrast, our study uses over 300,000 real omnichannel transactions from a multinational retailer, capturing both online and offline behavior across diverse geographic, cultural, and economic contexts.Furthermore, prior works rarely conduct loyalty prediction at both the individual and neighborhood levels simultaneously. Our dual-level modeling framework not only improves prediction accuracy but also enables practical CRM strategies tailored by customer segment and region, which is seldom addressed in existing empirical studies.This research addresses both theoretical and empirical gaps by combining sentiment analysis (via BERT), cultural dimensions (Hofstede), and economic indicators with advanced learning models such as GCN and Transformer. Unlike earlier studies that rely solely on RFM or CLV, our hybrid approach enhances prediction accuracy by modeling complex behaviors within a dynamic multichannel retail environment.

These variables can have a profound impact on customer purchase decisions, making it essential to simulate customer behavior more effectively by considering these factors. In addition to traditional models, this study employs advanced machine learning techniques, such as Graph Convolutional Networks (GCNs) and Transformer models, to analyze complex datasets and provide more precise loyalty predictions. These algorithms can simulate customer behavior with greater accuracy and offer optimized solutions for marketing strategies. To bridge the research gaps, an initial baseline model has been implemented to predict customer loyalty in a simplified manner. This baseline model excludes social, economic, and cultural variables, serving as a foundation for further model enhancements and refinements. This baseline model serves as a reference point for comparison with more advanced models that incorporate these additional factors. By combining these models, we can evaluate the impact of these factors on loyalty prediction accuracy and demonstrate how integrating them enhances prediction precision. In many cases, complex relationships exist within the stored organizational data, necessitating robust approaches to identify and analyze these intricate connections. While the use of machine learning techniques in predicting customer behavior and loyalty is increasing, this research gap presents new opportunities for applying advanced machine learning techniques such as Graph Convolutional Networks (GCNs), Transformer models. These techniques enable the analysis of complex datasets and more accurate customer loyalty predictions.

2-To Literature review the authors need to add some updated sources for Sentiment analysis, for BERT model, Evaluation Metrics and also for Hyperparameter Tuning, there are no sources.

Answer: We sincerely appreciate your highly valuable and pertinent feedback. Your requested suggestion has been graciously accepted and incorporated into the paper as follows:

In recent years, sentiment analysis has been enriched by deep learning techniques, particularly transformer-based models such as BERT (Bidirectional Encoder Representations from Transformers). BERT has demonstrated advanced performance in text classification tasks, including customer sentiment analysis in retail contexts

[134-136].

Evaluation metrics such as Accuracy, Precision, Recall, F1-Score, and Area Under the ROC Curve (ROC-AUC) are standard in classification problems and provide a comprehensive view of model performance

[137].

Hyperparameter tuning plays a critical role in optimizing model accuracy. Techniques such as Random Search and Particle Swarm Optimization (PSO) have been effectively applied to tune complex models in customer behavior studies [138-140].

Devlin, J., Chang, M. W., Lee, K., & Toutanova, K. (2019, June). Bert: Pre-training of deep bidirectional transformers for language understanding. In Proceedings of the 2019 conference of the North American chapter of the association for computational linguistics: human language technologies, volume 1 (long and short papers) (pp. 4171-4186).‏

Karabila, I., Darraz, N., EL-Ansari, A., Alami, N., & EL Mallahi, M. (2024). BERT-enhanced sentiment analysis for personalized e-commerce recommendations. Multimedia Tools and Applications, 83(19), 56463-56488

Geetha, M. P., & Renuka, D. K. (2021). Improving the performance of aspect based sentiment analysis using fine-tuned Bert Base Uncased model. International Journal of Intelligent Networks, 2, 64-69.‏

Sokolova, M., & Lapalme, G. (2009). A systematic analysis of performance measures for classification tasks. Information processing & management, 45(4), 427-437.‏

Lorenzo, P. R., Nalepa, J., Kawulok, M., Ramos, L. S., & Pastor, J. R. (2017, July). Particle swarm optimization for hyper-parameter selection in deep neural networks. In Proceedings of the genetic and evolutionary computation conference (pp. 481-488).‏

Soares, R. C., Silva, J. C., de Lucena Junior, J. A., Lima Filho, A. C., de Souza Ramos, J. G. G., & Brito, A. V. (2025). Integration of Bayesian optimization into hyperparameter tuning of the particle swarm optimization algorithm to enhance neural networks in bearing failure classification. Measurement, 242, 115829.‏

Agrawal, P. (2024, March). A survey on hyperparameter optimization of machine learning models. In 2024 2nd International Conference on Disruptive Technologies (ICDT) (pp. 11-15). IEEE.‏

3-The Methodology is well done and explained, but needs some improvements such as: for the regression equation determined by the authors must be added a clear explanation of the increasing/decreasing of the values for each independent variable and their influences on dependent variable how is explained (loyalty)

Answer: Thank you for your valuable feedback. The requested changes have been incorporated into the manuscript as follows:

The regression equation obtained via SEM indicates both the magnitude and direction of influence that each independent variable exerts on customer loyalty. The standardized path coefficients are interpreted as follows:

Customer Emotion (β = 0.26): A one-unit increase in the emotional sentiment score—extracted via BERT from customer reviews—is associated with a 0.26 standard deviation increase in the loyalty score. This highlights emotional engagement as the strongest driver of loyalty in our model.

Customer Lifetime Value (β = 0.21): Customers with higher CLV tend to be significantly more loyal. A one-unit increase in standardized CLV leads to a 0.21 increase in loyalty. This confirms the role of long-term customer profitability in shaping brand commitment.

RFM Score (β = 0.19): This composite score reflects recency, frequency, and monetary value. Frequent and recent buyers contribute positively to loyalty. Each one-unit rise in RFM predicts a 0.19 unit rise in loyalty.

Customer Segment (β = 0.16): Premium segments (as derived from purchasing patterns) show higher loyalty compared to new or occasional buyers.

Total Purchases (β = 0.12): Although positively correlated, this variable has a relatively smaller effect. It may overlap with CLV and RFM, suggesting diminishing marginal returns from purchase count alone.

Customer Churn (β = -0.06): This variable exhibits a negative relationship, indicating that an increase in churn probability decreases loyalty. Although the effect is modest, it is statistically significant and useful for early churn detection.

Together, these results offer a holistic view of loyalty as a function of emotional, behavioral, and financial dimensions. Notably, emotional factors (β = 0.26) have a stronger effect than pure purchase frequency (β = 0.12), highlighting the psychological aspect of loyalty in omnichannel contexts.

4-Theoretical implication need to be more developed and the Practical ones, written in italic both, to be easy perceived by the readers, to be more explained for the categories implied: not only for retailers, but also for customers and society/police makers in the field.

Answer: Thank you for your valuable feedback. The requested content has been added to the discussion and conclusion section as follows:

Practical Implementation for Retail Managers:

Personalized Customer Interaction: Retailers can categorize customers based on loyalty scores and tailor their marketing messages for each group.

Dynamic Pricing and Advertising: Real-time analysis of customer behavior enables retailers to adjust pricing and advertising strategies dynamically and purposefully.

Improved Customer Experience: Analyzing customer behavior patterns and sentiments helps address their concerns and improve the shopping experience.

Scenario-Based Decision-Making: By using reinforcement learning (Q-learning), businesses can simulate different economic and competitive scenarios and design effective strategies for retaining customers under various conditions.

Theoretically, this research contributes to the field by demonstrating the impact of combining socio-cultural, emotional, and economic indicators with traditional transactional features using an adaptive machine learning framework. It expands existing loyalty models by including neighborhood-level segmentation and dynamic sentiment analysis. From a practical perspective, the model offers key benefits for different stakeholders. Retailers, can optimize promotional strategies, inventory planning, and customer segmentation by using loyalty prediction scores. Customers,will experience improved personalization and satisfaction through adaptive recommendation systems informed by loyalty insights. Policy makers, can gain insights into socio-economic factors affecting consumer behavior, which can guide policy design in digital commerce and consumer protection. These stakeholder-focused benefits underline the model's relevance for both business application and policy development.

Therefore, integrating advanced machine learning techniques in predicting customer loyalty provides an innovative approach for multichannel retailers to strengthen customer loyalty and optimize performance. By combining social, economic, and behavioral data in predictive models, businesses can make data-driven decisions that enhance customer relationships, improve satisfaction, and ultimately increase profitability. Implementing these insights helps managers achieve sustainable growth in a competitive market and optimize their operational efficiency. Data analysis and the use of learning models always face challenges. In this study, due to the great diversity of data and the numerous features available, it is not unexpected that Emcom faced challenges such as adjusting and optimizing model parameters and preventing overfitting of the model. To deal with this problem, parameter tuning techniques such as PSO were used. This technique helped prevent overfitting and improved the performance of the model. Another challenge is the diversity and complexity of omnichannel data collected from different sales channels, such as physical and online stores. Therefore, great care must be taken in choosing a prediction model to increase the accuracy of the prediction. In this study, by using hybrid approaches, we have been able to achieve high accuracy in predicting loyalty. However, conducting this research comes with certain limitations. Some of these limitations include:

Reviewer 2:

1- The paper mentions using a Kaggle dataset, but doesn't fully describe its limitations. A detailed description of the dataset's characteristics (sample size, geographic distribution, temporal coverage, etc.) and a discussion of potential biases is needed. The claim of "diverse demographics, geographic locations, product categories, and brands" needs substantiation with specific numbers and details. The fact that the data comes from a single, unnamed omnichannel retailer significantly limits the generalizability of the findings. The authors acknowledge this limitation, but a more thorough discussion of the potential biases introduced by this single-source data is needed.

Answer: Thank you for your valuable feedback. The requested content has been added to the discussion and conclusion section as follows:

Although the statistics presented in Table 8 indicate an acceptable level of demographic and product diversity, several serious limitations must be considered:

Single-source bias: The data originates from a single multi-channel retai

---

## [Decision Letter · Decision Letter 1]

31 Jul 2025

Predicting Customer Loyalty in Omnichannel Retailing Using Purchase Behavior, Socio-Cultural Factors, and Learning Techniques

PONE-D-25-27280R1

Dear Dr. Shima Roosta,

We’re pleased to inform you that your manuscript has been judged scientifically suitable for publication and will be formally accepted for publication once it meets all outstanding technical requirements.

Kind regards,

Vincenzo Basile, PhD

Academic Editor

PLOS ONE

Additional Editor Comments:

Thank you for your resubmission and for the thoughtful revisions you have made to your manuscript titled “Predicting Customer Loyalty in Omnichannel Retailing Using Purchase Behavior, Socio-Cultural Factors, and Learning Techniques.”

After reviewing both referee reports, I am pleased to inform you that your manuscript is now **accepted for publication** in PLOS ONE.

Reviewer #1 confirms that all previous concerns have been satisfactorily addressed and supports the publication. Reviewer #2 acknowledges substantial improvements but raises several **additional suggestions** primarily related to structure, interpretative depth, and discussion of results. While these are relevant and valuable observations, they do not affect the core validity, methodology, or reproducibility of the study. As such, they are not considered essential for acceptance at this stage.

I encourage you, however, to take these points into account in your future work and in potential follow-up publications, particularly regarding:

Reducing redundancy in variable explanations;Deepening the interpretation of key predictive variables;Expanding on business implications of predictive outcomes.

Reviewers' comments:

Reviewer's Responses to Questions

**Comments to the Author**

Reviewer #1: All comments have been addressed

Reviewer #2: (No Response)

2. Is the manuscript technically sound, and do the data support the conclusions?

Reviewer #1: Yes

Reviewer #2: Yes

3. Has the statistical analysis been performed appropriately and rigorously?

Reviewer #1: Yes

Reviewer #2: No

4. Have the authors made all data underlying the findings in their manuscript fully available?

Reviewer #1: Yes

Reviewer #2: Yes

5. Is the manuscript presented in an intelligible fashion and written in standard English?

Reviewer #1: Yes

Reviewer #2: Yes

Reviewer #1: The paper entitled “Predicting Customer Loyalty in Omnichannel Retailing Using Purchase Behavior, Socio-Cultural Factors, and Learning Techniques” is interesting and apply a few models based on modelling and prediction.

To be publisehd the authors have made all the proposed improvements:

-In the Introduction, the authors added the text about the novelty of this paper, and the gap in the literature review and empirical comparing to their own study;

-To Literature review was imporoved by adding updated sources for Sentiment analysis, for BERT model, Evaluation Metrics and also for Hyperparameter Tuning;

-In the Methodology, for the regression equation were added clear explanatiosn of the increasing/decreasing of the values for each independent variable and their influences on dependent variable (loyalty).

-Theoretical implications and the Practical ones, were developed and better explained for the categories implied: not only for retailers, but also for customers and society/policy makers in the field.

Therefore, having these measures for improvements in view, and the improvements made by the authors, the paper receives acceptance of the paper to be published.

Reviewer #2: The paper has been sufficiently corrected but some concerns still need to be addressed in the revised manuscript.

1. There is redundancy in Explanation of Variables and Models. The same behavioral and socio-cultural variables (e.g., frequency, recency, education, income) are repeatedly described in multiple places across different model sections. This redundancy leads to unnecessary length and reduces reader engagement. The authors may consolidate the description of input variables in one section and refer to it in subsequent model analyses to improve clarity and brevity.

2. While some models indicate feature importance (e.g., Random Forest or Gradient Boosting), the rationale for why certain variables dominate is not critically interpreted. For example, the finding that "recency" or "education level" are dominant is stated but not discussed in the context of omnichannel consumer psychology or literature.

3. Models like Logistic Regression, KNN, Random Forest, and XGBoost are presented with results, but their comparative strengths and weaknesses are not clearly synthesized.

4. Although socio-cultural factors are part of the feature set, their contribution to model predictions is only numerically reported. There is minimal discussion on how cultural or demographic differences may drive customer loyalty patterns, especially across different segments.

5. There is no mention of class imbalance in the loyalty prediction task (e.g., loyal vs. non-loyal customers). If the dataset is imbalanced (which is common), accuracy can be misleading, and metrics like F1-score or AUC should be prioritized. Clarify whether the dataset is balanced and, if not, how this was handled (e.g., SMOTE, stratified sampling).

6. Across model results, accuracy is repeatedly highlighted, even though other metrics (recall, F1-score) are more relevant for churn/loyalty prediction. In practical CRM applications, identifying true loyal or non-loyal customers (recall/precision) matters more than overall accuracy. Reorient the evaluation around more appropriate performance metrics and discuss the implications of model bias or false positives.

7. The models predict loyalty, but there is no demonstration of how these predictions translate into improved business outcomes (e.g., increased retention, conversion). This makes it difficult to assess the practical value of the model for decision-making. Suggest or simulate a business application of the model (e.g., targeted promotion or personalized communication).

**Do you want your identity to be public for this peer review?** For information about this choice, including consent withdrawal, please see our Privacy Policy

Reviewer #1: **Yes: ** Florea Nicoleta-Valentina

Reviewer #2: **Yes: ** Dr. Rinku Sharma Dixit, PhD

---

## [Editor Report · Acceptance letter]

PONE-D-25-27280R1

PLOS ONE

Dear Dr. Roosta,

I'm pleased to inform you that your manuscript has been deemed suitable for publication in PLOS ONE. Congratulations! Your manuscript is now being handed over to our production team.

Kind regards,

on behalf of

Dr. Vincenzo Basile

Academic Editor

PLOS ONE